# Interpretable Neural ODEs for Gene Regulatory Network Discovery under Perturbations

Zaikang Lin [* 1 2 3 4]  Sei Chang [* 1 5]  Aaron Zweig [* 1 5 6]  Minseo Kang [1 7]  Fabian J. Theis [3 4]  Elham Azizi [8 6]
David A. Knowles [1 5]

## Abstract

Modern high-throughput biological datasets containing thousands of perturbations enable large-scale discovery of causal graphs that represent regulatory interactions between genes. Differentiable causal graphical models and regression-based methods have been developed to infer gene regulatory networks (GRNs) from interventional datasets. However, existing approaches fail to capture the non-linear dynamics of biological processes such as cellular differentiation. To address this limitation, we propose *PerturbODE*, a novel framework that employs interpretable neural ordinary differential equations (neural ODEs) to model cell state trajectories under perturbations and derive the underlying causal GRN from the neural ODE parameters, enabling downstream simulation of unseen genetic interventions. The GRN is encoded via a single-hidden-layer feed-forward network, implicitly grouping genes into interpretable co-regulated modules. We demonstrate PerturbODE's efficacy in GRN inference and extension to perturbation response prediction across both simulated and real overexpression datasets.

---

[*]Equal contribution  [1]New York Genome Center, New York, U.S. [2]Department of Applied Mathematics and Applied Physics, Columbia University, New York, U.S. [3]Institute of Computational Biology, Helmholtz Munich, Munich, Germany [4]Department of Mathematics, Technische Universität München, Munich, Germany [5]Department of Computer Science, Columbia University, New York, U.S. [6]Irving Institute of Cancer Dynamics, New York, U.S. [7]Department of Industrial Engineering and Operations Research, Columbia University, New York, U.S. [8]Department of Biomedical Engineering, Columbia University, New York, U.S. Correspondence to: Zaikang Lin <zl3135@columbia.edu>, David A. Knowles <dak2173@columbia.edu>.

*Proceedings of the 43rd International Conference on Machine Learning*, Seoul, South Korea. PMLR 306, 2026. Copyright 2026 by the author(s).

## 1. Introduction

GRNs capture the complex regulatory interactions between genes that dictate cell function, development, and responses to environmental changes. High-throughput perturbation assays with single-cell RNA sequencing (scRNA-seq) read-outs, such as Perturb-seq (Dixit et al., 2016) or open reading frame (ORF) overexpression (Joung et al., 2023), enable measurement of gene expression changes across cell types resulting from genetic perturbations. However, inferring GRNs from scRNA-seq experiments remains challenging due to the problem's exponential search space.

Regression-based approaches train a separate regression for each gene from all other genes, with random forests showing promising performance (e.g., GENIE3). While these classical approaches have performed well on several GRN recovery benchmarks (Pratapa et al., 2020; Huynh-Thu et al., 2010), they primarily capture statistical associations. Therefore, they struggle to infer the directionality of regulatory interactions and distinguish direct regulation from mediated effects. Moreover, they do not explicitly account for interventional responses.

In contrast, recent causal graphical models have been developed to leverage the increasing availability of perturbational datasets in single-cell genomics. These datasets enable causal inference by introducing targeted genetic interventions (e.g., gene knockouts or overexpression) and measuring the resulting transcriptional responses, thereby approximating do-interventions rather than relying on observational associations. Causal graphical methods explicitly encode the relationships between causal variables (genes) and simulate interventions (e.g., by removing edges). This enables them to leverage intervention information as inductive bias and generate samples from learned interventional distributions (Tejada-Lapuerta et al., 2025). These methods usually learn a directed acyclic graph (DAG) corresponding to the underlying GRN through a continuous, albeit non-convex, optimization program (Zheng et al., 2018; Fang et al., 2024; Brouillard et al., 2020; Lopez et al., 2022).

Causal graphical models have traditionally focused on learning graph structures from CRISPR-based gene knockdown

or overexpression Perturb-seq. These perturbations typically induce modest changes, slightly shifting cell state without driving transitions into distinct cell types. In contrast, new ORF overexpression single-cell experiments enable large perturbations, allowing us to explore a wider spectrum of cell states and regulations. Notably, the Transcription Factor (TF) Atlas applied single-cell resolution assays to systematically study the effects of overexpressing 1,836 TFs in embryonic stem cells, yielding more than 1.1 million cell profiles collected 7 days after perturbation (Joung et al., 2023). TFs, proteins that bind DNA to regulate gene expression, play a crucial role in defining cell states. Their overexpression can induce fate transitions that closely mimic natural development processes, thereby offering a means to model how TFs direct stem cells along trajectories toward diverse differentiated cell types, such as myocytes and neurons. However, causal graphical models lack an explicit representation of the underlying regulatory dynamics governing gene expression. Extensive fluorescence-based experiments have demonstrated that gene regulatory dynamics can be effectively modeled by non-linear dynamical systems (Alon, 2006; Setty et al., 2003; Kalir & Alon, 2004). Modeling these non-linear dynamics is essential for a more accurate representation of gene regulation under genetic interventions.

Lastly, many existing methods fail to account for the biologically important concept of gene modules. In complex eukaryotic cells, sets of functionally related genes exhibit highly correlated expression patterns across cell types and conditions. The "gene modules" (often referred to as pathways) are typically co-regulated by specific transcription factors. In addition to being biologically well-motivated, gene modules reduce computational and statistical complexity from learning $n^2$ variables to $2n \times d$ ($n$ = number of genes, $d$ = number of modules) and improve performance in network recovery (Segal et al., 2005). The inferred gene modules also enable biological interpretation, which we determine by assessing their overlap with gene sets from databases such as Gene Ontology (GO) (The Gene Ontology Consortium et al., 2026).

To address these limitations, we propose *PerturbODE*, a novel neural ODE-based framework that 1) explicitly encodes the GRN in its parameters, enabling simultaneous trajectory inference and GRN discovery, 2) maps cell states into a lower-dimensional "gene module" space analogously to causal representation learning (CRL) in Scholköpf et al. (2021), 3) allows explicit input of which gene(s) were perturbed, a feature uncommon in CRL approaches, 4) can model cycles and non-linear gene interactions, and 5) leverages causal relationships to predict the effects of unseen perturbations. Trained on the TF Atlas scRNA-seq data that capture the differentiation pathways of cells perturbed by overexpression of more than a thousand TFs, PerturbODE

enables scalable and interpretable discovery of the gene dependencies that drive cellular differentiation.

## 2. Related Work

**Causal graph discovery from genetic perturbations**. Structure learning of causal graphs has recently been applied to Perturb-seq interventional experiments to infer underlying GRNs. In these graphs, nodes represent genes, and directed edges correspond to putative causal regulatory relationships between genes. Since the number of possible DAGs increases exponentially with the number of nodes, classical causal graph discovery approaches cannot scale beyond a modest number of genes (typically 50-200). NO-TEARS (Zheng et al., 2018) introduced a continuous optimization objective via the trace exponential acyclicity constraint, significantly reducing problem complexity and enabling gradient descent-based structure learning. Subsequent extensions have further improved scalability. NO-TEARS-LR (Fang et al., 2024) incorporates a low-rank assumption to efficiently infer large, dense DAGs. DCDI (Brouillard et al., 2020) adapts the continuous optimization approach to interventional data while introducing a non-linear parameterization, but can only scale up to 50 dimensions in its original implementation using the trace exponential acyclicity constraint. DCDFG (Lopez et al., 2022) addresses this limitation by employing a low-rank factor graph representation and spectral radius acyclicity constraint.

**Neural ODEs for cell trajectory inference and modeling gene regulation.** Differential equation-based models have long been considered the gold standard for modeling gene regulation due to their fidelity to our understanding of true biophysical mechanisms (Alon, 2006). Neural ODEs allow flexible parameterization and efficient training with differentiable ODE integration-solvers (e.g., via the adjoint method), allowing tractable mechanistic modeling of dynamics given data (Chen et al., 2018). Neural ODEs and their stochastic variants have been applied to trajectory inference, where the continuous development of cellular states is mapped over time. Jackson et al. (2023) parameterize ODEs with recurrent neural networks (RNNs) to model dynamics before obtaining the coefficient of partial determination to represent the contribution of each TF. Hossain et al. (2024) incorporate kinetics using biological priors (e.g., using the Hill function) and explicitly encode the GRN as model parameters. However, both methods are designed for a single experimental setting with densely sampled data points along a pseudotime trajectory and cannot adequately handle the growing availability of perturbational datasets.

**Causal graph learning through stationary diffusion.** The recently proposed method Bicycle (Rohbeck et al., 2024) considers the GRN as the linear drift of a stable Ornstein-Uhlenbeck (OU) process, approximating the steady-state

distribution under each intervention induced by the OU process by solving the Lyapunov equation. Despite the novelty in methodology, Bicycle can handle only a hundred or so genes and the linear drift assumption constrains all predicted distributions to be Gaussian, which struggles to capture perturbations with more diverse cell fates.

**Key Limitations**. Despite recent improvements to network inference, causal graphical methods lack the expressivity to capture cellular dynamics and regulatory cycles at scale. Previous neural ODE-based approaches (Hossain et al., 2024; Jackson et al., 2023) infer GRNs from a single experimental condition and fail to handle multiple genetic perturbations. PerturbODE integrates causal structure learning with trajectory inference into a unified, biologically grounded, and scalable framework that accurately models cellular dynamics and infers the underlying GRN from thousands of perturbations.

## 3. Methods

Let $\mathcal{I} = \{I_0, I_1, \ldots, I_R\}$ represent a set of $R$ intervention regimes, with $I_0$ denoting the control regime (no intervention). The training dataset $\mathcal{D} = \{Y^{(r)}\}_{r=0}^R$ is a family of empirical distributions in the gene expression space, each corresponding to an intervention regime. $Y^{(r)} \in \mathbb{R}^{n_r \times d}$ represents the $d$-dimensional gene expression measurements of $n_r$ cells under the intervention regime $I_r$. $Y^{(0)}$, the empirical gene expression distribution under the control regime, is used as the unperturbed initial state from which we integrate our neural ODE function $f_r$ to predict the perturbation effect and final gene expression state under a given intervention.

### 3.1. Neural ODE formulation for overexpression with shift intervention

For any cell subjected to intervention $I_r \in \mathcal{I} \setminus \{I_0\}$, its cellular dynamics are described by the ODE,

$$\frac{\partial y^{(r)}(t)}{\partial t} = f_r(y^{(r)}(t)) = A\sigma(\alpha \circ (By^{(r)}(t) - \beta)) + \sum_{j \in I_r} s_{j,r} \cdot \delta_j - Wy^{(r)}(t), \quad (1)$$

where $y^{(r)}(t) \in \mathbb{R}^d$ represents the expression vector at time $t$ for a cell under intervention $I_r$.

This system encapsulates the interaction between genes through a Multi-Layer Perceptron (MLP) with a single hidden layer. Each neuron in the hidden layer is analogous to a gene module encapsulating co-regulated genes or biological pathways as outlined in Segal et al. (2005). Module-based regulatory network structures have been established in prior literature. A well-characterized example is the regulatory circuit of E. coli's flagella production (Macnab, 2003; Alon, 2006). In Appendix A.23, we illustrate how this structure

could be represented as a two-layer MLP.

The matrix $B \in \mathbb{R}^{l \times d}$ represents a linear transformation from $d$-dimensional gene expression $y^{(r)}(t)$ to a $l$-dimensional latent ("module") space. $B_{jm}$ is the signed effect of the $m$-th gene's expression on the $j$-th module.

The gene module signals are then non-linearly transformed after shift and scaling through the non-linear activation function $\sigma(\cdot) : \mathbb{R}^l \to \mathbb{R}^l$. We use the logistic sigmoid function for gene module activation $\sigma(\cdot)$ due to its equivalence (when modeling log expression) to the Hill function, which, following basic chemistry principles, represents the effect of TF concentration on target gene transcription rate (Alon, 2006). The vector $\beta \in \mathbb{R}^l$ is a strictly positive bias that shifts the activation threshold of the function $\sigma$ in each module. The vector $\alpha \in \mathbb{R}^l$ is a scaling factor that modulates the rate of activation through a Hadamard (i.e., element-wise) product ($\circ$) with the gene modules.

The module activations regulate downstream genes by combining linearly with those from other modules. The matrix $A \in \mathbb{R}^{d \times l}$ maps the $l$-dimensional latent vector back to the $d$-dimensional gene expression space. $A_{mj}$ represents the influence of the $j$-th module on the transcription rate of the $m$-th gene.

The interaction between genes mediated by modules encodes our estimate of the GRN matrix, $\mathbf{G} = A \operatorname{diag}(\alpha)B$. Conveniently, working with the lower-dimensional module space reduces our task from learning the full gene-to-gene matrix of size $d \times d$ (i.e., $d^2$ parameters) to learning two factorized graphs of size $d \times l$ (i.e. $2dl$ parameters).

The matrix $W \in \mathbb{R}^{d \times d}$ is diagonal with strictly positive entries such that $W_{ii} > 0$ is the decay rate for gene $i$. The decay component $-Wy^{(r)}(t)$ represents cellular RNA levels decreasing over time due to molecular decay and concentration dilution as cells grow and divide. Decay not only accurately models the regulatory biology but also encourages stability in the ODE system to prevent extreme levels of gene expression by creating a trapping region.

Interventions on the system are captured by the shift term $\delta_j = \mathbf{e}_j \in \mathbb{R}^d$, a standard basis vector corresponding to the induced overexpression of gene $j$ (which in our case is a TF). The vector $\mathbf{e}_j$ encodes a 1 in the $j^{th}$ entry and 0 in all other entries, allowing variable dynamics between cells with overexpression of different TFs. The scaling term $s_r = (s_{1,r}, s_{2,r}, ..., s_{d,r})^\top$ specifies the strength of each intervention on each gene. Importantly, each entry in $s_r$ is unique to a given intervention, while all other learned model parameters ($A$, $B$, $W$, $\alpha$, and $\beta$) are shared between all interventions.

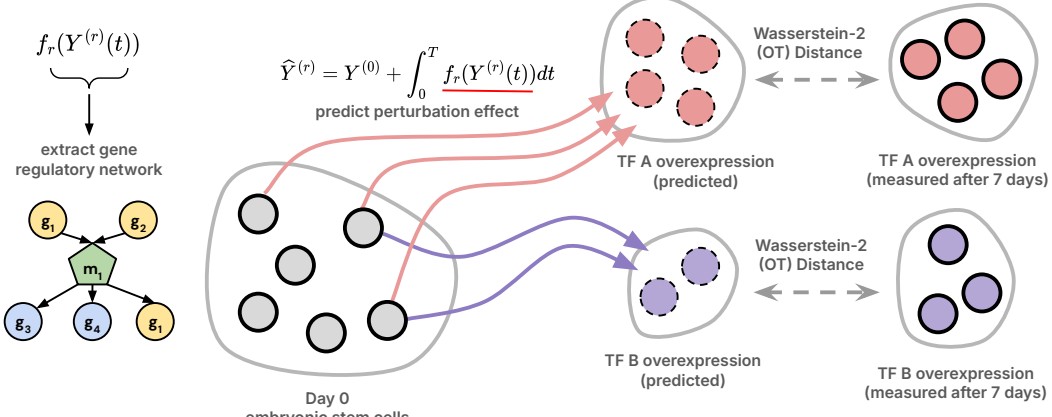

*Figure 1.* PerturbODE models the effect of a TF perturbation on stem cell differentiation by integrating the learned neural ODE function $f$ from the initial cell states $Y^{(0)}$ under intervention $I_r$. Under each perturbation (e.g. TF A overexpression), the predicted final cell states $\widehat{Y}^{(r)}$ are then compared to the observed differentiated gene expression values using the Wasserstein distance. From the parameters of $f$, we extract an underlying GRN.

### 3.2. Neural ODE formulation with perfect intervention

We adapt PerturbODE to model perfect interventions. Gene knockout or overexpression (CRISPR-a) under perfect intervention is modeled by removing the dependencies of the intervened genes on the parent nodes. In a system subject to a set $I_r$ of perfect interventions, the corresponding ODE is,

$$\frac{\partial y^{(r)}}{\partial t} = M_r A \sigma(\alpha \circ (B y^{(r)}(t) - \beta)) + \sum_{j \in I_r} s_j \cdot \delta_j - W y^{(r)}(t) \tag{2}$$

where $M_r = \mathbf{I} - \sum_{j \in I_r} \operatorname{diag}(\delta_j)$ is a masking matrix that removes the effect of other genes on the perturbed gene(s). For overexpression, $s_j > 0$ for all $j$, while for knockout, we set $s_j = 0$ for all $j$.

### 3.3. Mapping dynamics to targets using optimal transport

We train $f_r$ so that cells from $Y^{(0)}$ pushed forward through the dynamics fall close to $Y^{(r)}$. Specifically, we compute our target predictions $\widehat{Y}^{(r)}$ by numerically solving the ODE integration for each cell in the initial distribution,

$$\widehat{Y}^{(r)} = [\phi_T^r(y_1^{(0)}), \ldots, \phi_T^r(y_{n_r}^{(0)})]^\top$$
$$\phi_T^r(y_j^{(0)}) = y_j^{(0)} + \int_0^T f_r(y_j^{(r)}(t)) dt \tag{3}$$

where $j$ indexes cells in $Y^{(0)}$ and $\phi_T^r$ is the flow map of the ODE under intervention $I_r$ mapping initial cell state $y_j^{(0)}$ to its position at time $T$.

Given the lack of one-to-one correspondence between cells in the initial distribution $Y^{(0)}$ and samples in the target

distributions, we assess the quality of our predictions by measuring the Wasserstein-2 distance between observed distribution $Y^{(r)}$ and predicted distribution $\widehat{Y}^{(r)}$,

$$W_2(X, \widehat{X}) = \left( \min_{\Gamma \sim \Pi(X, \widehat{X})} \sum_{x,y} \|X_x - \widehat{X}_y\|_2^2 \Gamma_{xy} \right)^{1/2}, \tag{4}$$

where $\Pi$ represents the set of all optimal transport plans between each sample from data distributions $X$ and $\widehat{X}$, and $\Gamma$ represents the minimal-cost transport plan used to measure the dissimilarity between $X$ and $\widehat{X}$. The total loss function is defined as the average $W_2$ between $\widehat{Y}^{(r)}$ and $Y^{(r)}$ for all perturbations in $\mathcal{I}$ in addition to the $L_1$ norm of $B$ to encourage sparsity,

$$\mathcal{L}(\theta) = W_2(Y^{(r)}, \widehat{Y}^{(r)}) + \lambda |B|_1. \tag{5}$$

During training, for each intervention $I_r$, we push the control samples $Y^{(0)}$ through the map $\phi_T^r$ to obtain the predicted targets $\widehat{Y}^{(r)}$. We backpropagate through the loss and ODE solver to obtain gradients for all parameters. $L_1$ penalty is enforced only on $B$ because the network motif of a multiple-input feedforward loop is significantly less common than that of a multiple-output feedforward loop in known GRNs of yeast and E. coli (Kashtan et al., 2004). Training details and loss convergence plots can be found in Appendix A.8.

### 3.4. Diffusion-based regularization of neural dynamics

PerturbODE can optionally augment the primary training objective by using diffused target samples as alternative initial states. This additional regularization encodes our prior expectation that the final cell states should be locally stable,

helping to form a local contraction map that implies a locally stable fixed point, as ensured by the Contraction Mapping Theorem (Hunter & Nachtergaele, 2001). Interestingly, the stable fixed points establish the theoretical equivalence between PerturbODE and a deterministic structural causal model (SCM) (Mooij et al., 2013; Scholköpf et al., 2021).

The augmentation involves diffusing $Y^{(r)}$ using Brownian motion with a time step $\Delta t$ to generate diffused targets $Y^{(r)}_{\text{diff}}$. During a reduced time span $t \leq T$, $Y^{(r)}_{\text{diff}}$ is pushed forward through $\phi^{(r)}_t$ to obtain the predicted targets $\widehat{Y}^{(r)}_{\text{diff}}$, and we backpropagate against the augmented loss $\tilde{\mathcal{L}} = W_2(\widehat{Y}^{(r)}_{\text{diff}}, Y^{(r)}) + \lambda|B|_1$. During training, we alternate between using control samples $Y^{(0)}$ and diffused targets $Y^{(r)}_{\text{diff}}$ for each intervention. Information on diffusion training hyperparameters can be found in Appendix A.2.1.

## 4. Results: GRN Inference

For GRN inference benchmarks, we compare PerturbODE against the causal graph discovery methods DCDFG (Lopez et al., 2022), DCDI (Brouillard et al., 2020), NOTEARS (Zheng et al., 2018), NOTEARS-LR (Fang et al., 2024), and Bicycle (Rohbeck et al., 2024), as well as the random-forest-based method GENIE3 (Huynh-Thu et al., 2010), using both simulated data and large-scale perturbational scRNA-seq datasets.

### 4.1. GRN inference on simulated datasets

SERGIO (Dibaeinia & Sinha, 2020) and BEELINE (Pratapa et al., 2020) simulate single-cell gene expression data by sampling from a stochastic differential equation (SDE) parameterized by a user-provided ground-truth GRN. SERGIO simulates mature cells of any cell type in steady state or stem cells differentiating to multiple fates. It requires an acyclic input GRN and initializes cells at the mean of the steady-state distribution. BEELINE (BoolODE simulator) uses manually constructed cyclic GRNs with tens of genes that lead to convergent trajectories from predefined initial conditions.

We extended SERGIO and BEELINE to simulate gene expression with overexpression perturbations. We implemented interventions by masking the transcription induced by TF interactions (analogously to $M_r$ in Equation 2) of the intervened genes and adding a scalar to the intervened gene's transcription rate. We selected an experimentally curated GRN identified for yeast cells with dimension 400 as the input to SERGIO for simulation (Liu et al., 2015). The output synthetic dataset from SERGIO consists of 10,100 cells generated from 100 intervention schemes, each targeting 5 genes and one non-intervention (control) scheme. Each regime contains measurements of 100 cells. To evaluate the

models against a diverse range of networks, we simulated ten random DAGs with dimension 100 in the same manner. We used a bifurcating convergent synthetic cyclic network to simulate data using BEELINE. We compared the models' performance using the area under the precision-recall curve (AUPRC). Other metrics exhibit strong sensitivity to user-selected threshold values for edge classification, making them unreliable for benchmarking. Details on BEELINE, SERGIO, and the effects of thresholding and varying the number of modules are presented in Appendix A.22, A.21, and A.9, respectively.

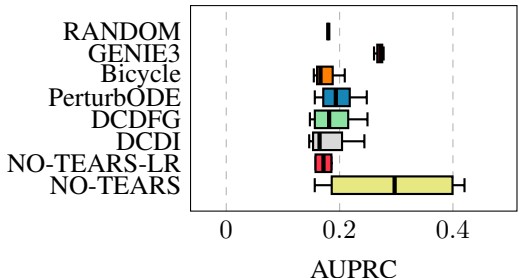

*Figure 2.* Performance metrics on data simulated from a synthetic GRN (10 genes) using BEELINE assuming perfect over-expression (CRISPR-a). 5 runs across different seeds and data splits are conducted for the synthetic GRN.

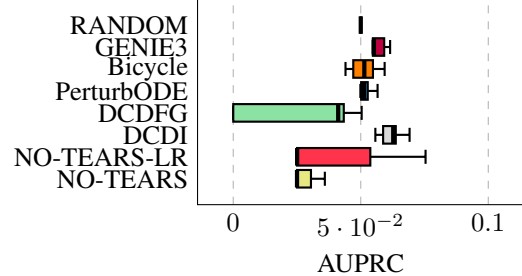

*(a)* 10 random acyclic GRNs (100 genes)

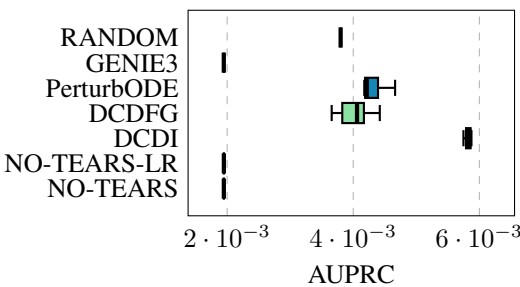

*(b)* Known yeast GRN (400 genes)

*Figure 3.* Performance metrics on SERGIO-simulated data assuming perfect over-expression (CRISPR-a). 5 runs across different seeds and data splits are conducted for the yeast GRN.

When scaling to simulations with 100 genes across 10 randomly generated GRNs (Figure 3a), PerturbODE and Bi-

cycle yield comparable performance, with DCDI and GE-NIE3 marginally outperforming both. DCDFG trails slightly behind PerturbODE and Bicycle, while NO-TEARS and NO-TEARS-LR show considerably reduced performance, highlighting their limitations in higher-dimensional regimes.

For simulations based on a known yeast GRN involving 400 genes (Figure 3b), PerturbODE exhibits greater comparative performance. Bicycle could not be assessed due to memory limitations, underscoring its lack of scalability. While DCDI achieves the highest performance in this setting, PerturbODE surpasses DCDFG, GENIE3, NO-TEARS, and NO-TEARS-LR. Notably, as the number of causal variables increases, other methods begin to break down due to memory constraints (Bicycle) or numerical instability during optimization (DCDI) resulting in NaN values appearing in training. In contrast, PerturbODE remains robust and consistent in its performance, which becomes more apparent in benchmarks on the TF Atlas dataset.

## 4.2. GRN inference on the TF Atlas

We trained PerturbODE on the TF Atlas to evaluate its performance on a large-scale real dataset. The TF Atlas overexpresses TFs and uses scRNA-seq to measure cell states after 7 days of perturbation (Joung et al., 2023). As this dataset maps the effects of TF overexpression, PerturbODE's inferred GRNs can uncover TF-to-gene interactions and network structure through TF modules.

Control samples (mCherry) were used as the initial gene expression states for solving the neural ODE (Equation 1). The resulting trajectories were used to predict final cell states, which were compared with ground-truth gene expression measurements obtained after 7 days of TF overexpression. Evaluation was performed on the union of the top 500 highly variable genes and the differentially expressed, experimentally intervened genes, yielding a total of 812 genes. Bicycle and DCDI were excluded from this comparison because they failed to run on datasets of this size due to memory constraints and numerical instability, respectively. PerturbODE was evaluated under both perfect (P) and imperfect (I) interventions, with PerturbODE* denoting the model variant that learns a tunable overexpression strength $s$ for each gene.

We consider two evaluation settings, using recall for the first and AUPRC for the second. For the recall-based evaluation, we benchmark the model on three well-characterized and experimentally validated human GRNs. These networks were constructed using RNA-seq and ATAC-seq data collected following thousands of transcription factor (TF) overexpression experiments in the TF Atlas study, together with previously CRISPR-validated GRN edges (Neijts et al., 2017; Krendl et al., 2017; Dejana et al., 2007) (see Appendix A.19). Since the GRNs contain only activating regulatory edges, our ground truth is restricted to true positives and

false negatives. Hence, we only computed recall scores for edge detection on the three TF Atlas human GRNs (Appendix A.9.1). To assess statistical significance of the model predictions, we compared the inferred GRNs with random matrices and computed a $p$-value (Appendix A.4).

**PerturbODE achieves the highest recall and the lowest p-value on the high-confidence TF Atlas human reference GRNs.** In Figure 4, PerturbODE* with imperfect intervention notably outperforms DCDFG, NO-TEARS, and NO-TEARS-LR in recall scores and p-values. While GENIE3 performs comparably, it lags behind PerturbODE overall. In Figure 5, we further assess the model's predictions across different sparsity levels by varying the thresholds for edge classification. PerturbODE* with imperfect interventions outperforms all other methods at most thresholds, with only GENIE3 marginally outperforming PerturbODE* at low sparsity levels.

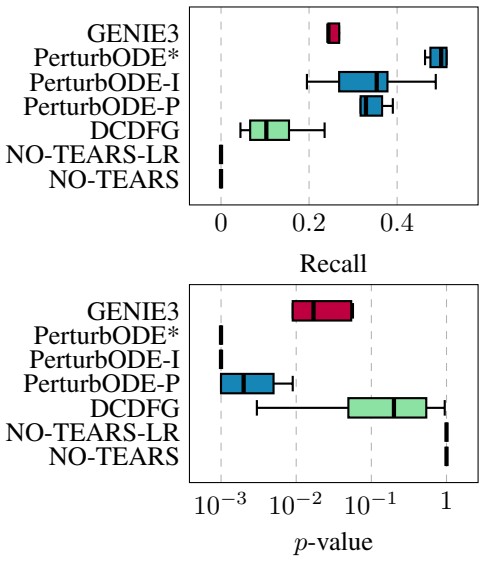

*Figure 4.* GRN inference performance on the TF Atlas dataset (812 genes), evaluated against three experimentally validated human GRNs. 5 runs across different seeds and data splits are conducted.

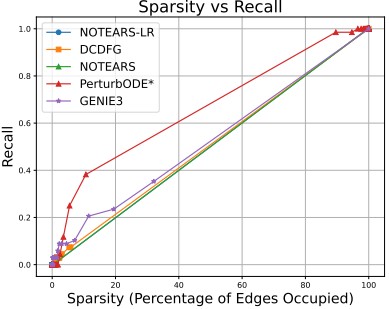

*Figure 5.* Recall across models on TF Atlas at various sparsity levels.

**PerturbODE achieves the highest AUPRC on reference GRNs from ChIP-Atlas.** To complement the preceding evaluation, which includes only validated activating regulatory edges, we additionally evaluate GRN recovery using a ChIP-seq-derived reference network of 79 genes associated with early embryonic differentiation, constructed from ChIP-Atlas data (Zou et al., 2024) (Figure 6). The selected genes are listed in Appendix A.11. Because the reference network is constructed from TF-binding assays, it is derived from measurements that can detect both the presence and absence of binding. Therefore, the network includes both positive and negative edges, enabling evaluation using AUPRC. Across all tested models, PerturbODE* achieves considerably higher AUPRC, reflecting PerturbODE*'s ability to recover the regulatory mechanisms specific to the differentiation trajectory. To demonstrate the consistency of our validation, we further benchmark another set of 41 genes annotated by the Gene Ontology Consortium as relevant to early embryonic development (The Gene Ontology Consortium et al., 2026) in Appendix A.12. Nevertheless, the performance of GENIE3 becomes comparable to PerturbODE* when we validate against a larger GRN (see Appendix A.13).

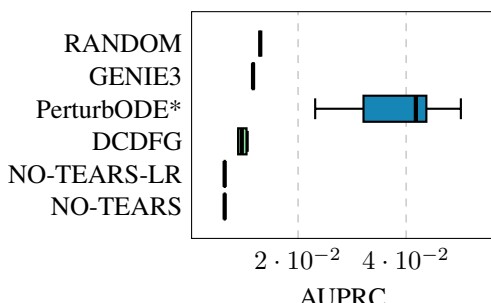

*Figure 6.* GRN inference on TF Atlas dataset (812 genes), validated with ChIP-Atlas (79 genes involved in early embryonic development). 5 runs (different seeds and data splits) are conducted.

In Table 1, we compare the False Discovery Rate (FDR) of each method at three threshold settings using the ChIP-Atlas reference GRN, which provides highly reliable ground truth for absent regulatory interactions. We computed the FDR using all 223 genes shared between TF Atlas and ChIP-Atlas to maximize the number of non-interaction edges. Compared to GENIE3, PerturbODE has a lower FDR at medium and high thresholds, while its FDR is slightly higher at the low threshold. PerturbODE comfortably outperforms DCDFG at all sparsity levels. The FDRs for NO-TEARS and NO-TEARS-LR could not be computed because both methods inferred nearly empty graphs over the set of genes present in ChIP-Atlas.

*Table 1.* Sparsity (%) and False Discovery Rate (FDR %) across three threshold settings (Low, Medium, High) on reference GRN derived from ChIP-Atlas.

| Method | Low | | Medium | | High | |
|---|---|---|---|---|---|---|
| | **Sparsity** | **FDR** | **Sparsity** | **FDR** | **Sparsity** | **FDR** |
| PerturbODE | 31.0 | 55.4 | 8.1 | **52.5** | 3.8 | **50.2** |
| GENIE3 | 33.5 | **54.1** | 8.2 | 54.1 | 3.9 | 55.0 |
| DCDFG | 38.7 | 58.8 | 8.3 | 58.8 | 3.7 | 58.7 |

**PerturbODE uncovers biologically meaningful gene network structure from its latent GRN modules.** We directly interpret PerturbODE's gene modules inferred from the TF Atlas dataset. Each module represents a set of gene-to-gene interactions derived from the model's $A$ and $B$ matrices (Equation 1), capturing upstream regulators and downstream targets. The most statistically significant modules recover known developmental GRNs, including those involved in anterior-posterior axis specification and trophoblast/vascular lineage induction. Our gene set enrichment analysis (GSEA) reveals a strong alignment of the modules with pathways involved in angiogenesis, fluid stress response, and early embryonic development. See Appendix A.20.2 for the full analysis.

## 5. Results: Perturbation Prediction

Predicting the effects of unseen interventions is a particularly challenging task. Here, we randomly select ten overexpressed TFs to be held out simultaneously during training. Note that the expression levels of these genes are observed, but their perturbations are not trained on. For this task, we compare PerturbODE with two linear SCMs (NO-TEARS and NO-TEARS-LR), a simple linear baseline introduced by Ahlmann-Eltze et al. (2025), the GNN-based method GEARS (Roohani et al., 2024), and the single-cell foundation model scGPT (Cui et al., 2024). DCDFG cannot sample cells given a learned GRN, and DCDI does not scale to this data. For linear SCMs, overexpression is implemented as an imperfect shift intervention by adding a bias to the mean of the distribution modeling the intervened nodes (see Appendix A.6).

We evaluate predictive performance through $W_2$ distance and cell type distance between the predicted and true distributions, complemented with visual inspection through low-dimensional UMAP embeddings (McInnes et al., 2018). Additional results evaluating predicted differentially expressed genes are presented in Appendix A.14. $W_2$ distance is calculated between the full distributions of the predicted and observed gene expressions. The cell type distance measures how closely our predictions align with the ground truth in estimating the cell type induced by each perturbation (see Appendix A.5). Figure 7 presents visualizations of model

predictions, with further examples in Appendix A.18.

*Table 2.* Predictive performance on held-out interventions in TF-Atlas.

| Method | $W_2$ | Cell Type Dist. |
|---|---|---|
| PerturbODE | $\underline{84 \pm 88}$ | $\mathbf{0.86 \pm 0.39}$ |
| GEARS | $\mathbf{65 \pm 7.3}$ | $0.89 \pm 0.20$ |
| scGPT | $91 \pm 14$ | $0.90 \pm 0.23$ |
| NO-TEARS-LR | $105 \pm 9$ | $1.08 \pm 0.26$ |
| Linear Baseline | $113 \pm 11$ | $\underline{0.88 \pm 0.20}$ |
| NO-TEARS | $396 \pm 9$ | $1.00 \pm 0.19$ |

*(a)* Predictive performance on 10 held-out interventions (median $\pm$ std).

| TF | PerturbODE | GEARS | scGPT | NO-TEARS-LR | Linear | NO-TEARS |
|---|---|---|---|---|---|---|
| ZNF69 | $\underline{85.4}$ | $\mathbf{63.3}$ | 88.1 | 106.0 | 110.9 | 164.9 |
| SETDB1 | 261.9 | $\mathbf{60.5}$ | $\underline{82.9}$ | 97.2 | 114.1 | 157.9 |
| POU2AF1 | 300.8 | $\mathbf{59.6}$ | $\underline{95.9}$ | 105.5 | 113.9 | 163.1 |
| ZBTB37 | $\underline{69.4}$ | $\mathbf{68.6}$ | 91.0 | 107.1 | 114.1 | 165.9 |
| IRF3 | $\underline{73.6}$ | $\mathbf{71.8}$ | 77.0 | 111.2 | 118.6 | 170.1 |
| ID1 | $\underline{79.6}$ | $\mathbf{69.2}$ | 82.9 | 109.7 | 114.7 | 168.7 |
| TEAD1 | 244.6 | $\mathbf{60.7}$ | 107.1 | 106.1 | $\underline{102.5}$ | 163.5 |
| ASCL1 | $\underline{94.1}$ | $\mathbf{84.2}$ | 116.4 | 134.8 | 144.2 | 192.7 |
| KCNIP4 | $\underline{82.7}$ | N/A | $\mathbf{63.4}$ | 104.7 | 112.6 | 163.7 |
| MSX2 | $\underline{66.7}$ | $\mathbf{65.1}$ | 91.2 | 103.7 | 110.2 | 164.6 |

*(b)* Test errors ($W_2$) for TF overexpressions across models.

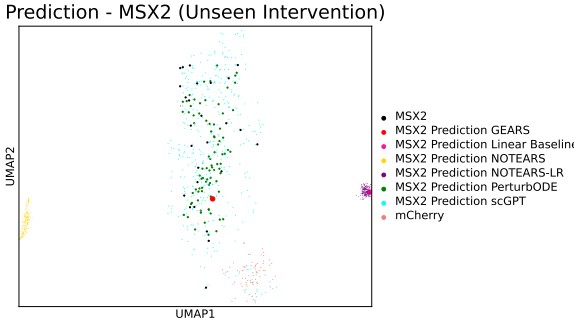

*(a)* Held-out TF (MSX2)

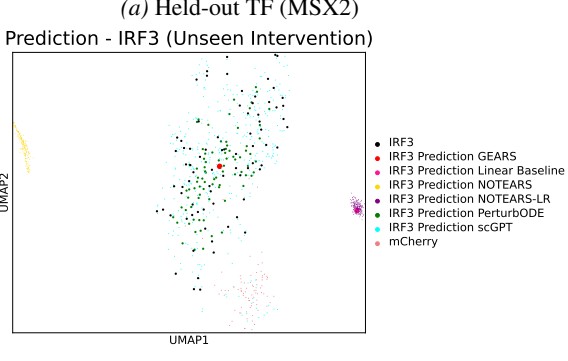

*(b)* Held-out TF (IRF3)

*Figure 7.* UMAP projections of cell embeddings: predictions for two held-out TFs—MSX2 and IRF3—found only in the test set.

PerturbODE substantially outperforms all linear methods in terms of $W_2$ distance and cell type distance with held-out interventions (Table 2a and Table 2b). However, when the model mispredicts, the associated prediction errors tend to be larger. PerturbODE outperforms scGPT in terms of $W_2$ distance and cell type distance. As illustrated by the UMAP visualization, scGPT predictions do not clearly capture the perturbation-specific shifts observed in the held-out data.

PerturbODE performs competitively with GEARS but falls short in terms of $W_2$ distance. Both methods achieve comparable performance in cell type classification, with PerturbODE showing a slight advantage. However, it is worth noting that GEARS integrates a knowledge graph from Gene Ontology (GO) annotations as an additional biological prior to aid its predictions. Augmenting our gene representations is a promising direction for future work.

# 6. Discussion

## 6.1. Identifiability

The question of identifiability of the ODE parameters is somewhat delicate. We assume the true underlying dynamics to be a deterministic system with an observed stationary distribution, and, in Appendix A.24, we provide an identifiability proof for our method. We additionally proved identifiability in a low-noise regime for an SDE extension of our model (Zweig et al., 2025).

## 6.2. Extension to Time Series Data

Future extensions of our work will include learning GRNs from longitudinal Perturb-seq data. Here we present preliminary results illustrating PerturbODE's ability to infer GRNs from a time-series Perturb-seq dataset (Ishikawa et al., 2023) (details in Appendix A.7). In Figure 8, we compare PerturbODE to GENIE3 and RENGE, a dynamical linear causal method developed for longitudinal Perturb-seq datasets (Ishikawa et al., 2023). The ground-truth GRN is obtained from the ChIP-Atlas. PerturbODE performs comparably to the other methods but exhibits substantially higher variability. Going forward, we aim to improve PerturbODE's stability and reduce variability.

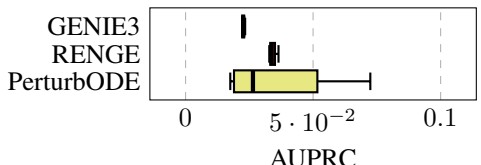

*Figure 8.* Performance metrics on longitudinal Perturb-seq (CRISPR-KO) data (103 genes), validated with ChIP-Atlas (31 genes). 5 runs (different seeds and data splits) are conducted.

# 7. Conclusion

PerturbODE is a scalable and biologically grounded causal approach to inferring GRNs from high-throughput genetic

perturbation data. Our method presents a dynamical alternative to traditional SCMs and regression-based methods. At its core, PerturbODE employs a two-layer neural network with sigmoid activation that mirrors cellular regulatory processes and gene modules. The framework offers both competitive predictive performance and biological interpretability of the learned parameters. In benchmarks of GRN inference using large-scale single-cell perturbation datasets, PerturbODE substantially outperforms existing methods. Crucially, we demonstrate the feasibility of using a GRN-driven approach to predict cellular responses to previously unseen perturbations. In future work, we aim to integrate ATAC-seq data to narrow down the candidate regulatory targets and mitigate false discoveries.

## Impact Statement

This work aims to accelerate biomedical research by helping to identify key genetic regulators and potential intervention targets for drug development. Any network or cell state predictions made using our method must be validated in the wet lab before using these results in downstream scientific or clinical decisions. In general, experimental validation will promote the responsible use of machine learning-based prediction models in biology.

## Acknowledgment

We thank Kui Ren, Won Eui Hong, Genevera Allen, Vladimir A. Kobzar, and Mingxuan Zhang for insightful discussions during the early stages of this work and on uncertainty quantification. We also thank the anonymous reviewers from ICLR 2025, ICML 2025, NeurIPS 2025, and ICML 2026 for their valuable comments and feedback.

This work was made possible by support from the MacMillan Family and the MacMillan Center for the Study of the Non-Coding Cancer Genome at the New York Genome Center.

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

# A. Appendix

## A.1. Preprocessing

The scRNA-seq gene expression matrix is normalized per cell by $10^4$ and $\log(1 + X)$ transformed. The total gene expression vector comprises RNA counts for $N$ genes consisting of all the TF overexpression genes $j$ and the top $k = 500$ highly variable genes.

For each TF gene $j$, we perform a Mann-Whitney U test on differential gene expression of TF $j$ between the unperturbed control samples in $X_0$ and overexpressed samples in $X_j$ consisting of $n_j$ cells. The returned p-value $p_j$ from the U test determines whether overexpression of the targeted TF gene $j$ is sufficiently induced in the experiments. The dataset is then filtered based on the criteria $\mathcal{D} = \{X_j \mid p_j < 0.1 \text{ and } n_j \geq 10, \ \forall j \in \{1, 2, \ldots, M\}\}$.

Overexpression distributions of the genes encoding the GRNs of interest are added to the training and validation dataset. In addition, when training for GRN inference only without trajectory prediction, distributions of TF overexpression encoded by the marker genes of the cell types or the developmental role targeted by the genes in the GRNs are included in the joint train, test, and validation dataset.

We design a train-test split based on TF overexpression genes to select $\mathcal{D}_{\text{train,val}}$ and $\mathcal{D}_{\text{test}}$. For each $X_j \in \mathcal{D}_{\text{train,val}}$ where $n_j \geq 100$, we apply a 80% to 20% training-validation split of the overexpression samples. If $n_j < 100$, we use all the samples in $X_j$ for $D_{\text{train}}$ due to an insufficient number of training samples.

Furthermore, we apply the **log1p** transformation to prevent negative predictions of gene expression and mitigate length biases in expression counts (Gorin & Pachter, 2023). This transformation results in a substantial improvement in model performance.

## A.2. Model Specifications

PerturbODE utilizes adaptive Runge-Kutta of order 5 of Dormand-Prince-Shampine which provides an exceptionally high order of accuracy and leverages its adaptive step size for efficient ODE solving. The adaptive step size also detects and handles a wide range of stiff ODEs. Differentiable numerical solution is computed via the adjoint method implemented in PyTorch by Chen (2021), available at `https://github.com/rtqichen/torchdiffeq`. The Sinkhorn-based $W_2$ distance is differentiable through the *GeomLoss* implementation in PyTorch (Feydy et al., 2019). For optimization, we used the `torch.optim.Adam` implementation of Adam (Kingma & Ba, 2015; Paszke et al., 2019) with a learning rate of $10^{-4}$.

For the GRN inference baseline methods, the authors of

DCDFG have implemented DCDI, DCDFG, NO-TEARS, and NO-TEARS-LR in the repository Lopez (2024), available at `https://github.com/Genentech/dcdfg`. Bicycle is implemented by Rohbeck et al. (2024) with code available at `https://github.com/PMBio/Bicycle`. GENIE3 is implemented by Huynh-Thu et al. (2010) with code available at `https://github.com/vahuynh/GENIE3`.

When benchmarking prediction of perturbation effects, we implemented the linear baseline of Ahlmann-Eltze et al. (2025) in numpy. GEARS is implemented by Roohani et al. (2024) at `https://github.com/snap-stanford/GEARS`, and scGPT is implemented by Cui et al. (2024) at `https://github.com/bowang-lab/scGPT`.

A.2.1. HYPERPARAMETERS

Spectral radius is used as the DAG constraint for DCDI, DCDFG, NO-TEARS, and NO-TEARS-LR. Notably, NO-TEARS and DCDI fail to run at dimensions higher than tens of variables with the trace exponential constraint. After hyperparameter-tuning, we set the optimizer learning rate to 0.001 and the regularization coefficient to 0.1. Other hyperparameters are set to their default values.

For Bicycle, the hyperparameters are chosen as follows: learning rate = 0.001, gradient_clip_var = 0.001, scale_kl = 1, scale_spectral = 0, and scale_lyapunov = 0.1. We set the other hyper-parameters as default.

When running GENIE3, we set $n\_trees = 1000$ after hyperparameter-tuning. Other hyperparameters were set to their default values.

For GEARS, after hyperparameter-tuning, we found the optimal hyperparameter to be the following: learning rate = 0.0001, $hidden\_size = 128$, and $epochs = 50$. Meanwhile, for scGPT, we found the optimal hyper-parameters to be the following: $embsize = 256$, $d\_hid = 512$, $n\_layers\_cls = 5$, $lr = 0.003$, and $batch\_size = 64$. The rest of hyper-parameters are set as default.

The number of modules is optimally set to 10 for NO-TEARS-LR and DCDFG. For PerturbODE, we set the number of modules to 100 for simulated data and 200 for TF Atlas. Details on performances across different number of modules in all models can be found in Figure 12.

As the number of modules increases, the model becomes closer to approximating the full graph. On the TF Atlas dataset, we demonstrate that the validation loss for PerturbODE decreases as the number of modules increases, plateauing after reaching 200 modules when training on TF Atlas (Fig. 9).

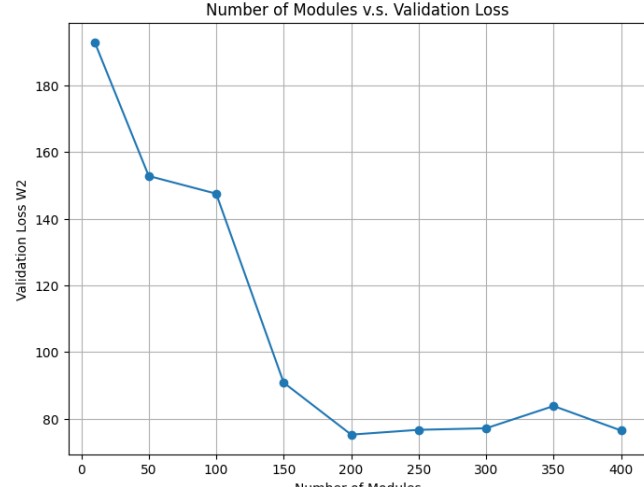

*Figure 9.* PerturbODE: number of modules v.s. validation loss in TF Atlas

We further evaluate GRN inference performance while varying the number of modules, benchmarking on all 223 genes shared between the TF Atlas and ChIP-Atlas. The optimal number of modules depends on a combination of universal approximation and latent factors. Performance improves as the number of modules increases, plateauing at 350 modules. For our evaluations in the main text, we selected 200 modules based on the $W_2$ validation loss for hyperparameter tuning.

*Table 3.* AUPRC values for different numbers of modules.

| Number of Modules | AUPRC |
| --- | --- |
| 10 | 0.017528 |
| 50 | 0.015840 |
| 100 | 0.018932 |
| 150 | 0.016924 |
| 200 | 0.017153 |
| 250 | 0.017880 |
| 300 | 0.019682 |
| 350 | 0.022517 |
| 400 | 0.021096 |

For both diffused and non-diffused training, PerturbODE uses 50 time steps when solving the ODE numerically. For diffused training, the time step duration $t$ is set to 0.1, while for non-diffused training, it is set to 25. The lasso regularization coefficient, $\lambda$, is set to 0.001. When computing the $W_2$ distance through Sinkhorn's algorithm, the coefficient for entropic regularization is set to 0.05. $\Delta t$ for the Brownian motion used to generate diffused data is set to 0.3.

*Table 4.* AUPRC comparison across numerical solvers, validated with ChIP-Atlas (all 223 genes available in the dataset).

| Numerical Method | AUPRC |
|---|---|
| dopri5 (adaptive) | 0.017 |
| rk4 | 0.017 |
| euler | 0.018 |

We further validate the performance of PerturbODE, without tunable perturbation strength, in GRN recovery with the TF Atlas dataset when different numerical methods are employed in Table 4. The model performance is consistent across different numerical methods.

Moreover, we showcase our model performance across various learn-rates (1 run for each learn-rate). The inference performance is consistent when there are no Out of Memory (OOM) errors.

*Table 5.* Effect of the learning rate on AUPRC. OOM indicates out-of-memory failure.

| Learning Rate | AUPRC |
|---|---|
| 1 | OOM |
| $10^{-1}$ | OOM |
| $10^{-2}$ | 0.021 |
| $10^{-3}$ | 0.021 |
| $10^{-4}$ | 0.022 |

### A.3. Runtime & Memory Comparison

In Table 6, we compare the runtime and memory consumption on SERGIO simulated data (400 genes) and TF Atlas (812 genes) across PerturbODE, DCDFG, and GENIE3. We evaluated all methods on a CPU because DCDFG's torch version does not support our GPU, and GENIE3 could only run on a CPU.

*Table 6.* Runtime and peak memory usage across datasets and GRN inference methods.

| Dataset | Method | Peak Memory | Runtime |
|---|---|---|---|
| TF Atlas | PerturbODE | 13.10 GB | > 1 day |
| | DCDFG | 0.93 GB | 4.85 hrs |
| | GENIE3 | 4.14 GB | > 1 day |
| SERGIO | PerturbODE | 3.7 GB | 12.8 hrs |
| | DCDFG | 0.93 GB | 0.32 hrs |
| | GENIE3 | 1.23 GB | 1.2 hrs |

### A.4. Comparison to Erdős-Rényi Random Graphs

We generate $10,000$ random graphs with the same density as our inferred GRN to numerically simulate the test statistics under Erdős-Rényi random matrices. The p-value is calculated using the equation,

$$p\text{-value} = \frac{1 + \#\{\tau^* \geq \tau\}}{1 + \Pi} \tag{6}$$

where $\tau$ is the test statistic, $\Pi$ indicates the total number of random graphs, and $\tau^*$ denotes the test statistics computed from each graph. The p-value quantifies how often a test statistic is observed (or a more extreme one) purely by chance.

When evaluating SERGIO simulated data, the test statistics used is the F1 score, whereas recall score is used for TF Atlas due to availability of only activator benchmark edges. To identify gene modules, we use test statistics based on the count of incoming edges to the module and outgoing edges from the module that are consistent with known regulatory relationships. Further, to identify the network motif of negative auto-regulation, the test statistic is the number of negative self-loops.

### A.5. Cell Type Distance

We provide a new metric to assess the accuracy of cell type prediction. We first average the gene expression values across all cells and classify the cell type of the averaged expression using k-nearest neighbor search ($k = 10$) in a 50-dimensional PCA space. The classification result produces a categorical vector $c$ representing the probabilities of each cell type. Our cell type distance is then defined as the $L_1$ distance $||\hat{c} - c||_1$, where $\hat{c}$ is the classification vector for our predicted distribution and $c$ is the vector for the observed distribution. A smaller distance indicates a better alignment of the cell type between our predictions and ground truth.

### A.6. Sampling from Linear SCMs for TF Atlas

For a learned GRN represented by $\mathbf{W}$ (ensured to be a DAG, or thresholded to enforce acyclicity), we sample from linear structural causal models (SCMs) using the following procedure. First, for each parent gene $i$ (master regulator) in the GRN, if not overexpressed, its expression level $X_i$ is sampled from a normal distribution, $X_i \sim \mathcal{N}(\mu, \sigma)$, where $\mu$ and $\sigma$ represent the mean and standard deviation of gene expression levels across all genes and cells in the TF Atlas, respectively. If $X_i$ is overexpressed, it is instead sampled from $X_i \sim \mathcal{N}(\mu_\gamma, \sigma_\gamma)$ where $\mu_\gamma$ and $\sigma_\gamma$ are the mean and standard deviation of gene expression levels in overexpression genes across all overexpressed cells.

Downstream genes are realized in Equation 7:

$$X_i = \sum_{X_j \in \text{pa}(X_i, \mathbf{W})} \mathbf{W}_{j,i} X_j$$

if $X_i$ is not overexpressed,

$$X_i = \sum_{X_j \in \text{pa}(X_i, \mathbf{W})} \mathbf{W}_{j,i} X_j + \gamma_i,$$

$$\gamma_i \sim \mathcal{N}(\mu_\gamma - \mu, \sigma_{\Delta\gamma})$$

if $X_i$ is overexpressed,

$$(7)$$

where $\sigma_{\Delta\gamma}$ is the standard deviation of the differences between overexpressed genes and mean expression levels (average over genes) across all overexpressed cells. Further, $\text{pa}(X_i, \mathbf{W})$ denotes all the parent genes (regulators) of gene $i$ in the GRN $\mathbf{W}$.

### A.7. Training on Time-series Perturb-seq Data

The RENGE dataset comprises time-resolved single-cell RNA-seq data capturing the developmental trajectories of human induced pluripotent stem cells (hiPSCs) across four time points following 23 CRISPR-based single-gene knockouts targeting key transcription factors in the pluripotency network (Ishikawa et al., 2023). We focus the analysis on 103 genes, including the 23 knockout genes and the 80 transcription factors showing the largest expression changes after gene knockout.

Instead of iteratively giving the model a perturbation before conducting backpropagation, we include 5 randomly chosen perturbations within each batch. Further, we backpropagate through the gradient of the aggregated $W_2$ distance between predicted distributions and target distributions across all time points in addition to the $L_1$ penalty term.

### A.8. PerturbODE Model Training

After training, the average $W_2$ distance on both the training and held-out validation datasets decreases notably and converges. The convergence rate of the $W_2$ distance varies for each TF in the training and validation sets.

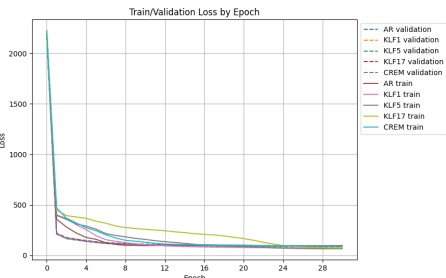

*Figure 10.* Convergence of $W_2$ losses for trajectory predictions of training and validation samples per TF. Average validation loss on TF Atlas is 78.88.

## A.9. Additional Thresholded Results

### A.9.1. THRESHOLDS

The performance evaluation in the main body mainly looks at AUPRC as the choice of thresholds could affect evaluation drastically. For additional evaluations, here we apply thresholds $\epsilon$ to the predicted GRN matrices $\mathbf{G}$, where any edge with a weight below $\epsilon$ is set to 0 and any edge whose weight exceeds $\epsilon$ is set to 1.

As recommended by their authors, DCDFG determines the threshold $\epsilon$ through binary search, using depth of 20 evaluations of an exact acyclicity test to find the largest possible DAG for each method. NO-TEARS and NO-TEARS-LR's $\epsilon$ are chosen to be 0.3 while DCDI's is set to 0.5 as recommended by the respective authors. For DCDI, NO-TEARS and NO-TEARS-LR different thresholdings such as binary search are attempted without meaningful change to the result. Different fixed values for $\epsilon$ were also experimented for DCDFG without improvements. The author of Bicycle did not disclose the appropriate threshold. We found the threshold of 0.005 to be the only one yielding reasonable results.

PerturbODE's $\epsilon$ threshold is determined using the formula $\epsilon = c \cdot \sigma$, where $\sigma$ represents the standard deviation of the inferred GRN matrix $\mathbf{G}$ across all entries, and $c$ is a positive scalar. For SERGIO simulated data with 400 genes, c = 1, and for that with 100 genes, c = 0.1. For TF Atlas, we set $c = 0.1$.

As the authors of GENIE3 did not recommend a default threshold, we choose a threshold for GENIE3 in the same manner as PerturbODE. For SERGIO simulated data with 400 genes, $c = 5$, and for that with 100 genes, $c = 0.1$. Lastly, for TF Atlas, $c = 1$.

In simulated data, $c$ is chosen so that the number of predicted edges approximately matches the number of edges in the ground truth GRN. And for TF Atlas, $c$ is chosen so that GENIE3 and PerturbODE predict a similar number of edges.

In SERGIO simulated data, we used ground truth networks that contain only activator edges. For the TF Atlas, the literature-curated GRN edges consist only of activator edges. PerturbODE and Bicycle can distinguish activators and repressors, whereas DCDI and DCDFG only identify edge existence. When evaluating PerturbODE and Bicycle on SERGIO simulated data, we treat incorrect sign as a false positive.

BEELINE simulated data contains both activator and repressor edges. Hence, we take the absolute values over reference GRN and predicted GRN before comparison.

### A.9.2. THRESHOLDED PERFORMANCE

PerturbODE demonstrates higher precision, recall, and F1 scores compared to DCDFG, NO-TEARS, and NO-TEARS-LR, while performing comparably to DCDI in these metrics (Fig. 11b, Fig. 11a). DCDI is the state-of-the-art method that outperforms PerturbODE in lower dimensional simulated datasets ($100 - 400$ genes), but it lacks scalability. In fact, for dimensions greater than 400, DCDI simply fails to execute, even with the more computationally feasible spectral radius acyclicity constraint. Details of the performance across all models with varying numbers of modules are provided in A.10. PerturbODE's main contribution is its ability to train on real datasets with thousands of genes, while maintaining competitive predictive performance.

For evaluation, we threshold the weights of the output GRNs to obtain classification metrics (details in Appendix A.9.1). To further address the discrepancies between graph sparsity and predictive performance, we employed random graphs to generate an empirical null for each test statistic for random graphs with the same edge density. We compare the precision-recall test statistics of the predicted GRN against those from $10,000$ Erdős-Rényi random networks, yielding empirical $p$-values (for details, see Appendix A.4).

There is considerable variation in recall scores for PerturbODE especially in the simulated yeast dataset. This is likely due to the high sparsity in the ground truth GRN, which leads to weak signals in the simulated dataset. This results in false negatives. Further, $L_1$ penalty is enforced on the individual matrix. As multiplication of sparse matrices is not always sparse, the number of predicted edges tends to fluctuate. Denser predictions would have higher recall scores.

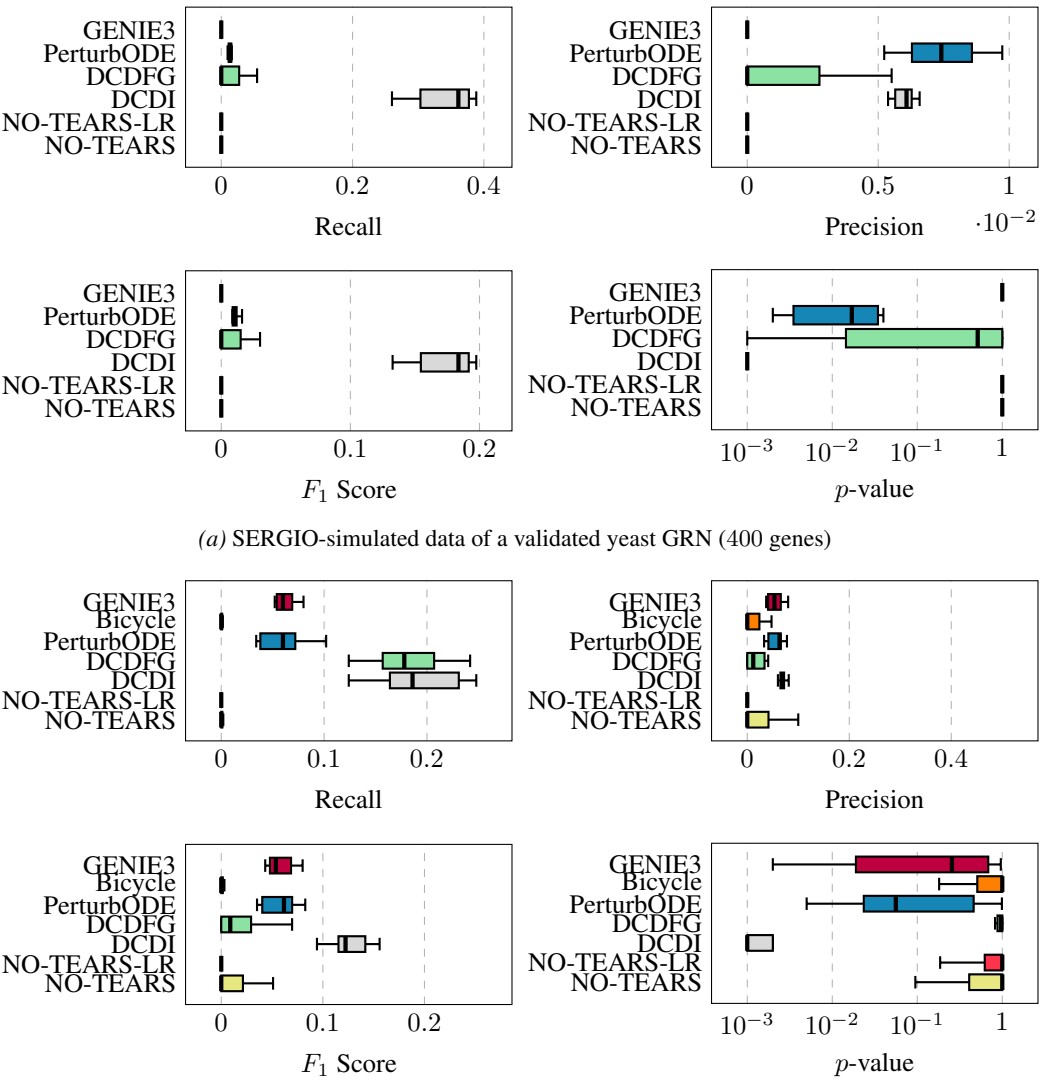

*(a)* SERGIO-simulated data of a validated yeast GRN (400 genes)

*(b)* SERGIO-simulated data of 10 random acyclic GRNs (100 genes)

*Figure 11.* Performance metrics on SERGIO-simulated data, assuming perfect intervention by overexpression (CRISPR-a).

*Table 7.* Average number of edges predicted by all methods across datasets

| METHOD | GROUND TRUTH | PERTURBODE | NO-TEARS | NO-TEARS-LR | DCDI | DCDFG | GENIE3 |
|---|---|---|---|---|---|---|---|
| YEAST GRN ($dim = 400$) | 623 | 1205.8 | 0.0 | 0.0 | 24332.8 | 4293.8 | 721.6 |
| RANDOM DAGS ($dim = 100$) | 500 | 552.0 | 0.0 | 7.1 | 1423.7 | 215.1 | 586.2 |
| TF ATLAS ($dim = 812$) | N/A | 101404.2 | 438.0 | 76.0 | N/A | 72884.0 | 93430.0 |

Table 7 presents the number of edges predicted by each model across different datasets using the aforementioned thresholds.

### A.9.3. MEAN AND STANDARD DEVIATION OF RESULTS

*Table 8.* Mean and standard deviation across models for yeast simulated by SERGIO

| Method | Recall | | Precision | | AUPRC | | F1 | | p-value | |
|---|---|---|---|---|---|---|---|---|---|---|
| | Mean | Std | Mean | Std | Mean | Std | Mean | Std | Mean | Std |
| GENIE3 | 0.0000 | 0.0000 | 0.0000 | 0.0000 | 0.0019 | 0.0000 | 0.0000 | 0.0000 | 1.0000 | 0.0000 |
| PerturbODE | 0.0151 | 0.0043 | 0.0079 | 0.0017 | 0.0045 | 0.0002 | 0.0115 | 0.0028 | 0.0458 | 0.0554 |
| DCDFG | 0.0315 | 0.0414 | 0.0026 | 0.0032 | 0.0041 | 0.0003 | 0.0170 | 0.0223 | 0.6058 | 0.4829 |
| NO-TEARS-LR | 0.0000 | 0.0000 | 0.0000 | 0.0000 | 0.0027 | 0.0015 | 0.0000 | 0.0000 | 1.0000 | 0.0000 |
| NO-TEARS | 0.0000 | 0.0000 | 0.0000 | 0.0000 | 0.0019 | 0.0000 | 0.0000 | 0.0000 | 1.0000 | 0.0000 |
| DCDI | 0.3499 | 0.0470 | 0.0061 | 0.0004 | 0.0059 | 0.0001 | 0.1780 | 0.0237 | 0.0010 | 0.0000 |

*Table 9.* Mean and standard deviation across models for random DAGs simulated by SERGIO

| Method | Recall | | Precision | | AUPRC | | F1 | | p-value | |
|---|---|---|---|---|---|---|---|---|---|---|
| | Mean | Std | Mean | Std | Mean | Std | Mean | Std | Mean | Std |
| GENIE3 | 0.0640 | 0.0098 | 0.0565 | 0.0143 | 0.0575 | 0.0058 | 0.0596 | 0.0117 | 0.4109 | 0.3712 |
| DCDI | 0.1960 | 0.0387 | 0.0695 | 0.0084 | 0.1528 | 0.0195 | 0.1327 | 0.0204 | 0.0145 | 0.0320 |
| NO-TEARS-LR | 0.0006 | 0.0013 | 0.0300 | 0.0605 | 0.0427 | 0.0190 | 0.0153 | 0.0308 | 0.8430 | 0.3144 |
| DCDFG | 0.0164 | 0.0188 | 0.0247 | 0.0251 | 0.0307 | 0.0203 | 0.0206 | 0.0214 | 0.8675 | 0.2923 |
| PerturbODE | 0.0622 | 0.0225 | 0.0579 | 0.0147 | 0.0521 | 0.0020 | 0.0601 | 0.0155 | 0.3039 | 0.3628 |
| NO-TEARS | 0.0006 | 0.0009 | 0.0683 | 0.1484 | 0.0530 | 0.0747 | 0.0345 | 0.0745 | 0.7911 | 0.3292 |

*Table 10.* Mean and standard deviation across models for TF Atlas

| Method | Recall | | p-value | |
|---|---|---|---|---|
| | Mean | Std | Mean | Std |
| GENIE3 | 0.2634 | 0.0284 | 0.0172 | 0.0182 |
| NO-TEARS | 0.0000 | 0.0000 | 1.0000 | 0.0000 |
| NO-TEARS-LR | 0.0000 | 0.0000 | 1.0000 | 0.0000 |
| DCDFG | 0.1353 | 0.0692 | 0.4158 | 0.3692 |
| PerturbODE (imperfect interv) | 0.3659 | 0.0556 | 0.0042 | 0.0032 |
| PerturbODE* (imperfect interv) | 0.4976 | 0.0195 | 0.0010 | 0.0000 |
| PerturbODE (perfect interv) | 0.3561 | 0.0946 | 0.0236 | 0.0452 |

### A.10. GRN Inference Results with Different Number of Modules in Benchmarks

PerturbODE and NO-TEARS-LR maintain consistent performance across different numbers of modules, while DCDFG achieves its best results with 10 modules. Figures 12 and 13 illustrate the performance of all models across varying numbers of modules in the SERGIO and TF Atlas datasets. Here, the threshold for PerturbODE is set at $c = 0.1$.

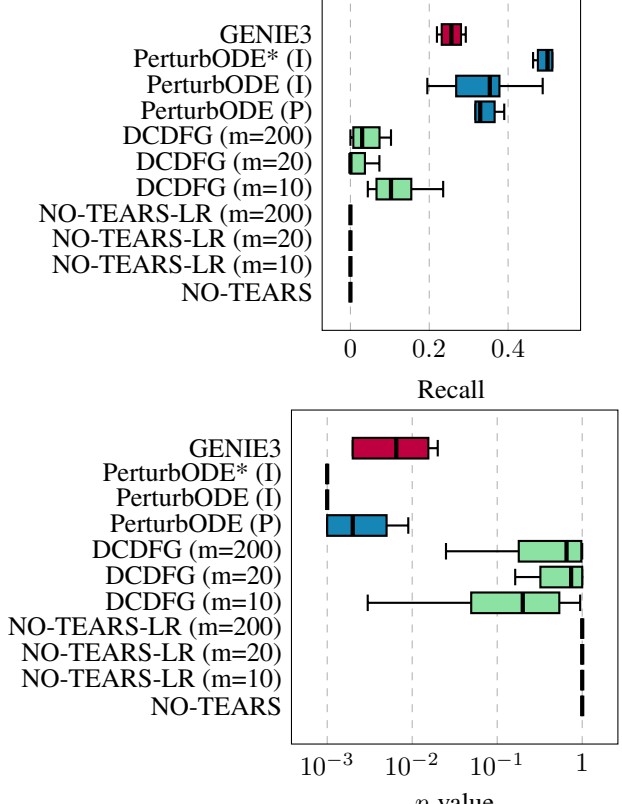

*Figure 13.* GRN Inference on TF Atlas Dataset (812 genes). Models with different numbers of modules are compared.

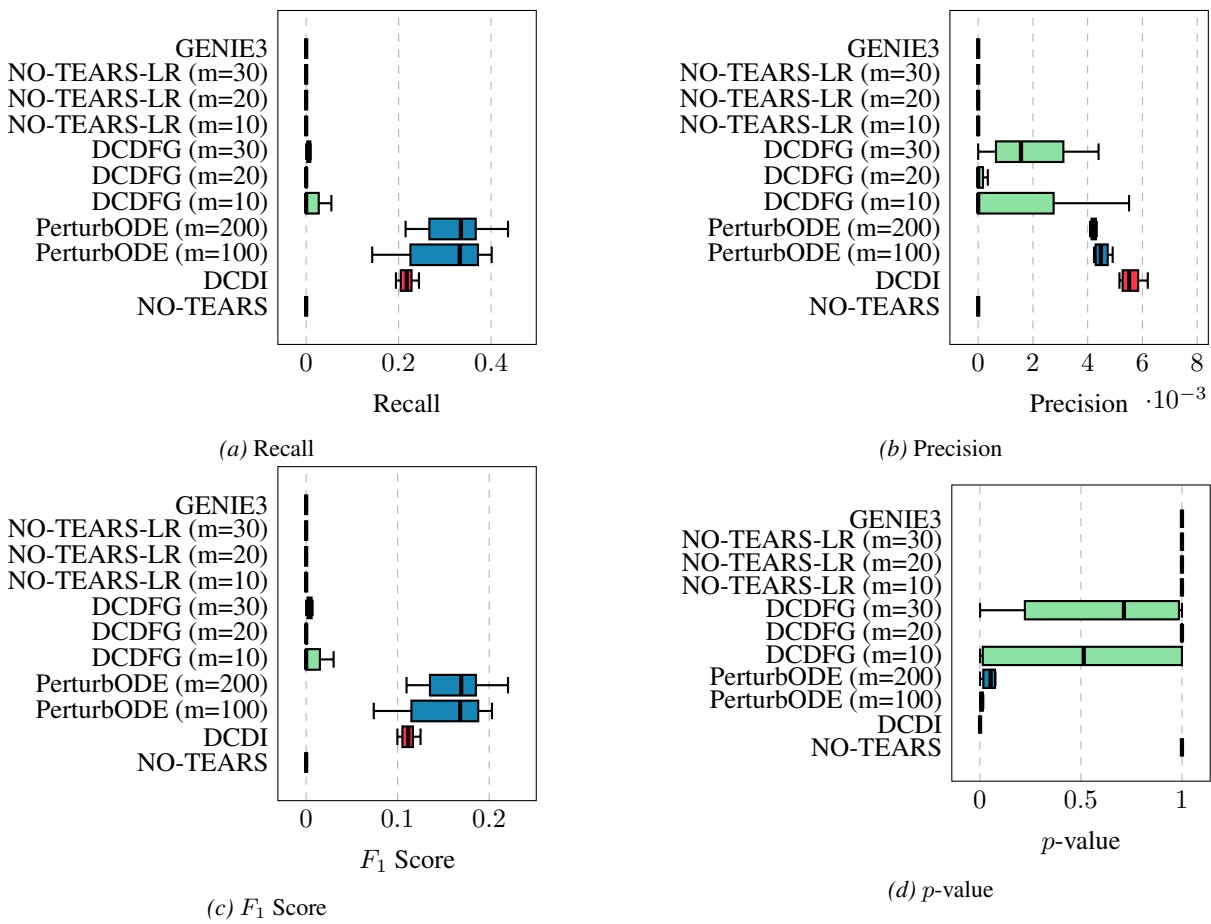

*(a)* Recall

*(b)* Precision

*(c)* $F_1$ Score

*(d)* $p$-value

*Figure 12.* Perfect intervention overexpression (CRISPR-a) SERGIO simulation GRN inference. Ground truth GRN is a known yeast GRN (400 genes). Models with different number of modules are compared in their $F_1$ score and $p$-value.

## A.11. TF Atlas: Genes Relevant to Early Embryonic Development

For GRN validation against ChIP-Atlas, we analyzed the subset of genes shared between the TF Atlas and ChIP-Atlas datasets. This set includes transcription factors and regulatory genes with well-established roles in early embryonic development, such as anterior–posterior patterning, germ-layer specification, early lineage commitment, and morphogenesis.

The analyzed gene set comprised the following: HOXA6, HOXA9, HOXA11, HOXB1, HOXB6, HOXB7, HOXB8, HOXC5, HOXC9, HOXC11, HOXC13, HOXD1, HOXD4, HOXD9, HOXD11, HOXD13, CDX1, CDX2, CDX4, EOMES, MESP1, MEIS1, MEIS2, PBX1, PBX3, ID1, ID3, LIN28B, HMGA1, SATB1, GATA3, GATA4, TBX2, TBX3, PRRX1, PITX1, NKX2-1, ISL1, WT1, NR2F2, NR5A1, NR5A2, HNF4A, HNF4G, HNF1B, PDX1, ASCL1, PAX2, PAX5, PAX6, PAX7, PAX8, EMX1, CRX, LHX5, LHX6, MSX2, ZFHX4, GRHL1, GRHL3, OVOL1, OVOL3, TFAP2A, TFAP2C, TEAD1–4, YAP1, TAL1, FLI1, ETV2, LMO1, LMO3, RUNX1T1.

The selected genes include canonical HOX and CDX genes involved in axial patterning (HOXA6, HOXA9, HOXA11, HOXB1, HOXB6, HOXB7, HOXB8, HOXC5, HOXC9, HOXC11, HOXC13, HOXD1, HOXD4, HOXD9, HOXD11, HOXD13, CDX1, CDX2, CDX4; Neijts et al., 2017).

The intersecting gene set also contains developmental regulators involved in early embryonic lineage specification, cell-fate determination, and the formation of diverse mesodermal and ectodermal derivatives. EOMES and MESP1 are crucial in the specification of cardiac mesoderm during gastrulation (Costello et al., 2011). MEIS1, MEIS2, and PBX1 are essential for hematopoietic development and vascular patterning (Azcoitia et al., 2005). PBX3 is essential in organogenesis (Di Giacomo et al., 2006). ID1 and ID3 are expressed during murine gastrulation and neurogenesis (Jen et al., 1997). LIN28B plays an important role in germ layer specification in Xenopus (Faas et al., 2013). HMGA1 (Gandhi et al., 2020) plays a crucial role in neural crest induction. SATB1 regulates cell fate in early mouse embryo (Goolam & Zernicka-Goetz, 2017).

Transcription factors that regulate diverse organogenesis programs during embryonic development are also included. Knocking down GATA3 in primate embryos prevents trophectoderm specification (Krendl et al., 2017). GATA4 is involved in endoderm specification downstream of nodal signaling, while TBX3 promotes liver bud growth and differentiation (Zorn & Wells, 2009). TBX2 is important in early heart development (Harrelson et al., 2004). MHox, also known as PRRX1, is important in mesenchymal and skeletal organogenesis (Martin et al., 1995). PITX1 is crucial in

hindlimb morphogenesis (Szeto et al., 1999). NKX2-1 plays a significant role in lung organogenesis (Minoo et al., 1999). ISL1 is important in cardiac progenitor development (Cai et al., 2003). WT-1 is required for early kidney development (Kreidberg et al., 1993). HNF1B, PDX1, and HNF4A are crucial in pancreatic specification in embryonic endoderm (Gere-Becker et al., 2018; Ng et al., 2024). COUP-TFII, also known as NR2F2, is important in angiogenesis and cardiac development (Pereira et al., 1999). NR5A2 is required for primitive-streak morphogenesis (Labelle-Dumais et al., 2006). NR5A1 is essential for adrenal and gonadal organogenesis (Luo et al., 1994). HNF4G acts redundantly with HNF4A during maturation of the fetal intestine (Chen et al., 2019).

The shared gene set further encompasses transcription factors involved in embryonic neural development and differentiation. ASCL1 promotes neuronal differentiation of progenitors, while PAX6 promotes oligodendrogenesis (Guillemot, 2007). PAX2, PAX8, and PAX5 are important in midbrain and cerebellum development (Urbánek et al., 1997). Further, specification of the neural crest occurs during gastrulation and requires PAX7 (Basch et al., 2006). MSX2 is important in cranial neural crest development (Ishii et al., 2005). EMX1 regulates neuronal differentiation (Bishop et al., 2003). LHX5 is a key factor in posterior hypothalamic specification and hippocampal morphogenesis (Miquelajáuregui et al., 2015; Zhao et al., 1999). CRX regulates photoreceptor differentiation (Furukawa et al., 1997). LHX6 activity is required for normal migration and specification of cortical interneuron subtypes (Liodis et al., 2007). ZFHX4 is crucial for human brain development and neuronal differentiation (Pérez Baca et al., 2025). LMO1 and LMO3 play roles in neuronal development (Biswas et al., 2023; Hinks et al., 1997; Boehm et al., 1991).

Furthermore, we assessed TFs that regulate epithelial lineage specification and morphogenetic patterning during embryonic development. GRHL1 and GRHL3 targeted TFAP2C and TEAD family TFs to induce trophoblasts (Joung et al., 2023; Krendl et al., 2017). OVOL1 regulates the growth arrest of embryonic epidermal progenitor cells (Nair et al., 2006). TFAP2A regulates epithelial and neural crest developmental programs (Zhang et al., 1996). YAP1 controls epithelial progenitor patterning and compartment boundary formation during lung organogenesis (Mahoney et al., 2014).

Lastly, the gene set included early hemato-endothelial and vascular developmental regulators. FLI1 targeted AP-1 family TFs and ETV2 to induce vascular endothelial cells (Joung et al., 2023; Dejana et al., 2007). RUNX1T1 regulates endothelial angiogenesis and embryonic vascular development (Liao et al., 2017). TAL1 is important in developmental hematopoiesis (Hoang et al., 2016).

### A.12. TF Atlas: ChIP-Atlas Validation on Early Embryonic Development Genes from Gene Ontology Consortium (GO)

In Figure 14, we further validate our method using ChIP-Atlas on a set of 41 genes annotated by the Gene Ontology Consortium as relevant to early embryonic development (The Gene Ontology Consortium et al., 2026).

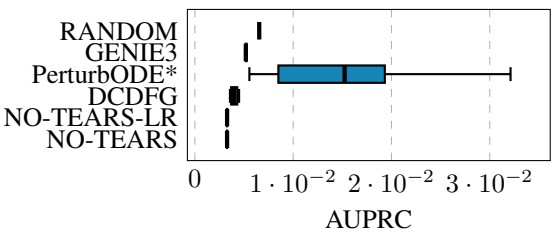

*Figure 14.* GRN inference on TF Atlas dataset (812 genes), validated with ChIP-Atlas on 41 genes involved in early embryonic development based on Gene Ontology (GO). 5 runs (different seeds and data splits) are conducted.

To define a set of genes involved in early embryonic development, we selected Gene Ontology (GO) biological process terms capturing major developmental events from early patterning to lineage specification and early organogenesis. Specifically, we included terms related to general embryo development and spatial patterning, including embryo development (GO:0009790), embryonic pattern specification (GO:0009880), regionalization (GO:0003002), anterior/posterior pattern specification (GO:0009952), and dorsal/ventral pattern formation (GO:0009953). We additionally included terms describing gastrulation and germ-layer establishment, namely gastrulation (GO:0007369), formation of primary germ layer (GO:0001704), cell fate commitment involved in formation of primary germ layer (GO:0060795), ectoderm formation (GO:0001705), endoderm formation (GO:0001706), and mesoderm formation (GO:0001707), as well as subsequent germ-layer developmental programs: ectoderm development (GO:0007398), endoderm development (GO:0007492), mesoderm development (GO:0007498), and mesodermal cell fate commitment (GO:0001710). Finally, to capture early axial, neural, hematopoietic, and cardiac developmental processes, we included somitogenesis (GO:0001756), neural tube formation (GO:0001841), neural crest formation (GO:0014029), embryonic heart tube development (GO:0035050), and embryonic hemopoiesis (GO:0035162).

### A.13. TF Atlas: ChIP-Atlas Validation on Full Shared Gene Set

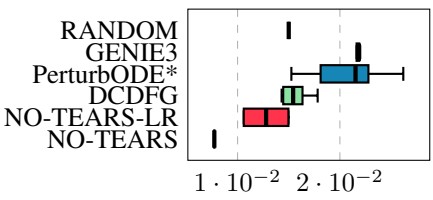

*Figure 15.* GRN inference on TF Atlas dataset (812 genes), validated against ChIP-Atlas on all 223 shared genes). 5 runs (different seeds and data splits) are conducted.

### A.14. Perturbation Prediction: Differentially Expressed Genes

We further report the F1 score of predicted up-regulated differentially expressed genes (DEGs) versus true up-regulated DEGs. For scGPT, NO-TEARS, NO-TEARS-LR, and PerturbODE, we computed DEGs using the Wilcoxon test in Table 11. In contrast, GEARS and the Linear Baseline generate only a single predicted sample; therefore, DEGs were derived by ranking genes based on log fold changes against pseudobulked control cell population. To ensure a fair comparison with GEARS and the Linear Baseline, we additionally pseudobulked PerturbODE's predictions and report the corresponding results in Table 12.

*Table 11.* F1 scores computed from the top 50 differentially expressed genes (DEGs) identified using the Wilcoxon test, considering only upregulated genes.

| Perturbation | NOTEARS | NOTEARS-LR | PerturbODE | scGPT |
|---|---|---|---|---|
| ASCL1 | 0.06 | 0.08 | **0.14** | 0.12 |
| ID1 | 0.02 | 0.00 | **0.04** | **0.04** |
| IRF3 | 0.02 | 0.00 | **0.04** | 0.02 |
| KCNIP4 | 0.02 | 0.00 | **0.06** | 0.02 |
| MSX2 | 0.02 | 0.00 | **0.06** | 0.04 |
| POU2AF1 | 0.04 | 0.02 | **0.06** | **0.06** |
| SETDB1 | **0.10** | 0.04 | 0.02 | 0.06 |
| TEAD1 | **0.06** | 0.00 | 0.02 | 0.04 |
| ZBTB37 | 0.02 | 0.04 | **0.12** | 0.06 |
| ZNF69 | **0.06** | 0.00 | **0.06** | **0.06** |
| **Median** | 0.03 | 0.00 | **0.06** | 0.05 |
| **Std.** | 0.026 | 0.025 | 0.037 | 0.029 |

*Table 12.* F1 scores computed from the top 50 DEGs obtained from delta-gene rankings, considering only upregulated genes. Predictions from PerturbODE are pseudobulked to enable comparison.

| Perturbation | GEARS | Linear Baseline | PerturbODE |
|---|---|---|---|
| ASCL1 | **0.10** | 0.00 | 0.08 |
| ID1 | 0.22 | 0.02 | **0.34** |
| IRF3 | **0.36** | 0.02 | 0.34 |
| KCNIP4 | – | 0.04 | **0.38** |
| MSX2 | 0.10 | 0.02 | **0.28** |
| POU2AF1 | 0.26 | 0.00 | **0.30** |
| SETDB1 | 0.04 | 0.04 | **0.20** |
| TEAD1 | 0.24 | 0.08 | **0.30** |
| ZBTB37 | 0.22 | 0.04 | **0.30** |
| ZNF69 | **0.36** | 0.08 | 0.34 |
| **Median** | 0.22 | 0.03 | **0.30** |
| **Std.** | 0.107 | 0.027 | 0.091 |

## A.15. Ablation Study & Power Analysis

Ablation study and power analysis were performed for PerturbODE* trained on TF Atlas.

### A.15.1. POWER ANALYSIS

Figure 16 shows the number of perturbations included for training plotted against recall and p-value. The overall trend shows that as the number of perturbations grows, recall increases and p-value decreases.

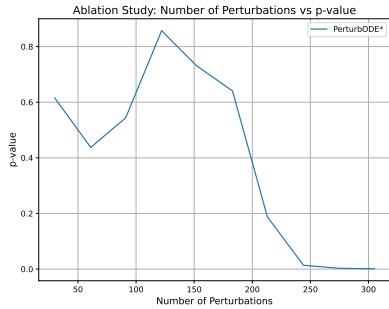

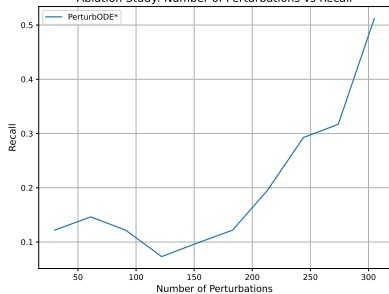

*Figure 16.* Ablation study: TF Atlas number of perturbations v.s. recall and p-value.

### A.15.2. ABLATION STUDY: L1 PENALTY

Figure 17 shows the change in recall and p-value when varying the $L_1$ penalty coefficient for $B$. The ablation study shows that PerturbODE* yields statistically significant results when $\lambda \leq 0.001$. Further, it is evident that as $\lambda$ increases above $0.01$, the number of edges predicted increases again. Our GRN is encoded as $\mathbf{G} = A \operatorname{diag}(\alpha) B$. The multiplication of sparse matrices is not necessarily sparse. Further analysis shows strong penalization of $B$ leads to overly dense $A$, as the model resorts to $A$ for data fitting. This could lead to an increase in the number of edges predicted.

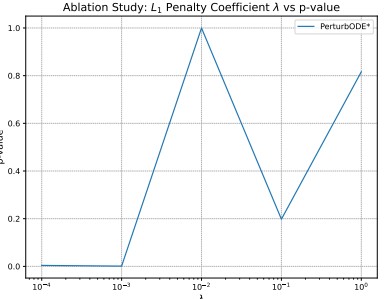

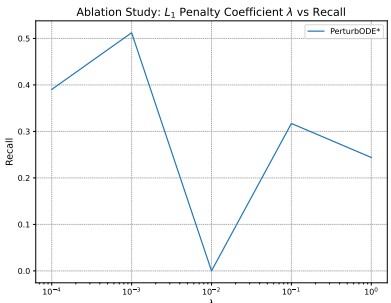

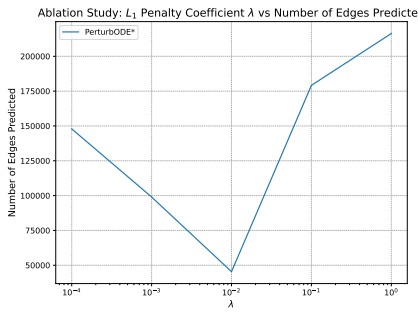

*Figure 17.* Ablation study: TF Atlas $L_1$ penalty coefficient $\lambda$ v.s. recall, p-value, and number of edges predicted.

We also report an ablation study by the removal of L1 penalty term below (benchmarking on all 223 genes in the ChIP-Atlas):

*Table 13.* Effect of the L1 penalty on AUPRC across five runs.

| Setting | Mean AUPRC | Std. |
|---|---|---|
| Without L1 Penalty | 0.017 | 0.001 |
| With L1 Penalty | 0.022 | 0.004 |

### A.15.3. ABLATION STUDY: WASSERSTEIN LOSS

We report the MSE between the mean of the predicted and true distributions (i.e. the L2 norm on the mean difference). We give this loss below benchmarking on all 223 genes in the ChIP-Atlas.

*Table 14.* Effect of the loss function on AUPRC across five runs.

| Loss Function | Mean AUPRC | Std. |
|---|---|---|
| L2 (Pseudobulk MSE) | 0.0158 | 0.001 |
| Wasserstein-2 | 0.022 | 0.004 |

### A.15.4. ABLATION STUDY: SALIENCY MAP

We empirically explored extracting GRNs via saliency map via a 3-layer MLP. The results are worse than our standard approach. Comparison can be seen below.

*Table 15.* Effect of model architecture and GRN inference strategy on AUPRC across five runs.

| Model | Mean AUPRC | Std. |
|---|---|---|
| saliency map | 0.0151 | 0.001 |
| standard | 0.022 | 0.004 |

### A.16. Statistical Inference: Generalizability and Stability Analysis

For stability analysis, we bootstrapped (sampled with replacement) the TF Atlas dataset 105 times to evaluate consistency in the edges selected by PerturbODE. We also filtered the list of TF perturbations that PerturbODE trains on down to the TFs pertinent to the ground truth GRNs in order to reduce training time. Then the gene expression space is the union between the filtered TF list and the top 50 highly variable genes, resulting in 52 genes. For generalizability analysis, we performed a similar procedure but using 133 different train-validation splits. Train-validation split is chosen to be $80\% - 20\%$, where, for each interventional distribution, $20\%$ of samples are withheld for the validation set. The validation set is used as stopping criterion (a hyper-parameter) for training.

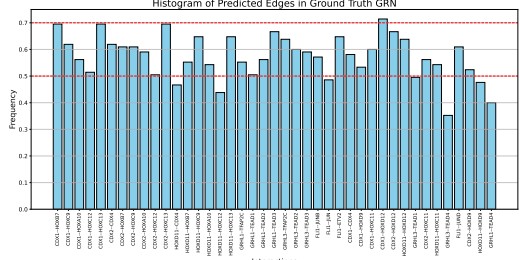

*(a)* Stability Analysis: Ground Truth Edges Selected by PerturbODE.

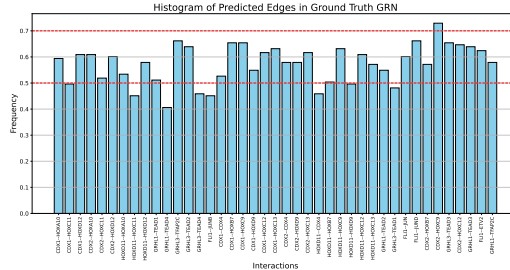

*(b)* Generalizability Analysis: Ground Truth Edges Selected by PerturbODE.

*Figure 18.* Comparison of Stability and Generalizability Analyses for Ground Truth Edges Selected by PerturbODE.

Figures 18a and 18b indicate that PerturbODE selects the ground truth edges roughly $50\%$ to $70\%$ of the time in both the stability and generalizability analyses. While a highly consistent model would ideally surpass a $75\%$ selection rate, these results nonetheless reflect a reasonable degree of reliability given the inherent complexity of the task. Future enhancements to the model may further improve this consistency.

### A.17. Prediction on Unseen Intervention: All UMAP and PCA Plots

Figures 19, 20, show the detailed results on predictions on test data (unseen intervention) through UMAP and PCA.

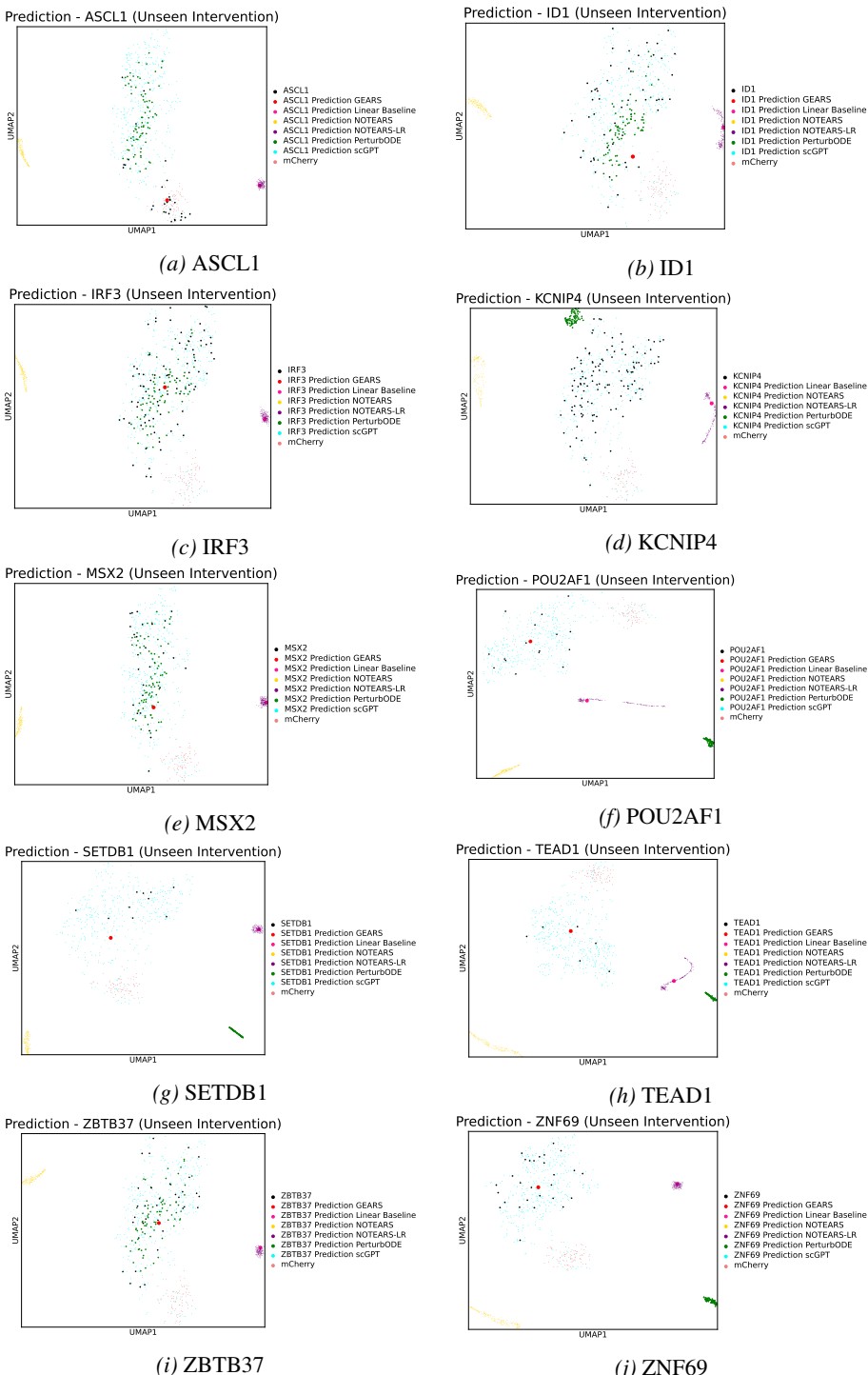

*Figure 19.* UMAP of predictions on unseen interventions across models.

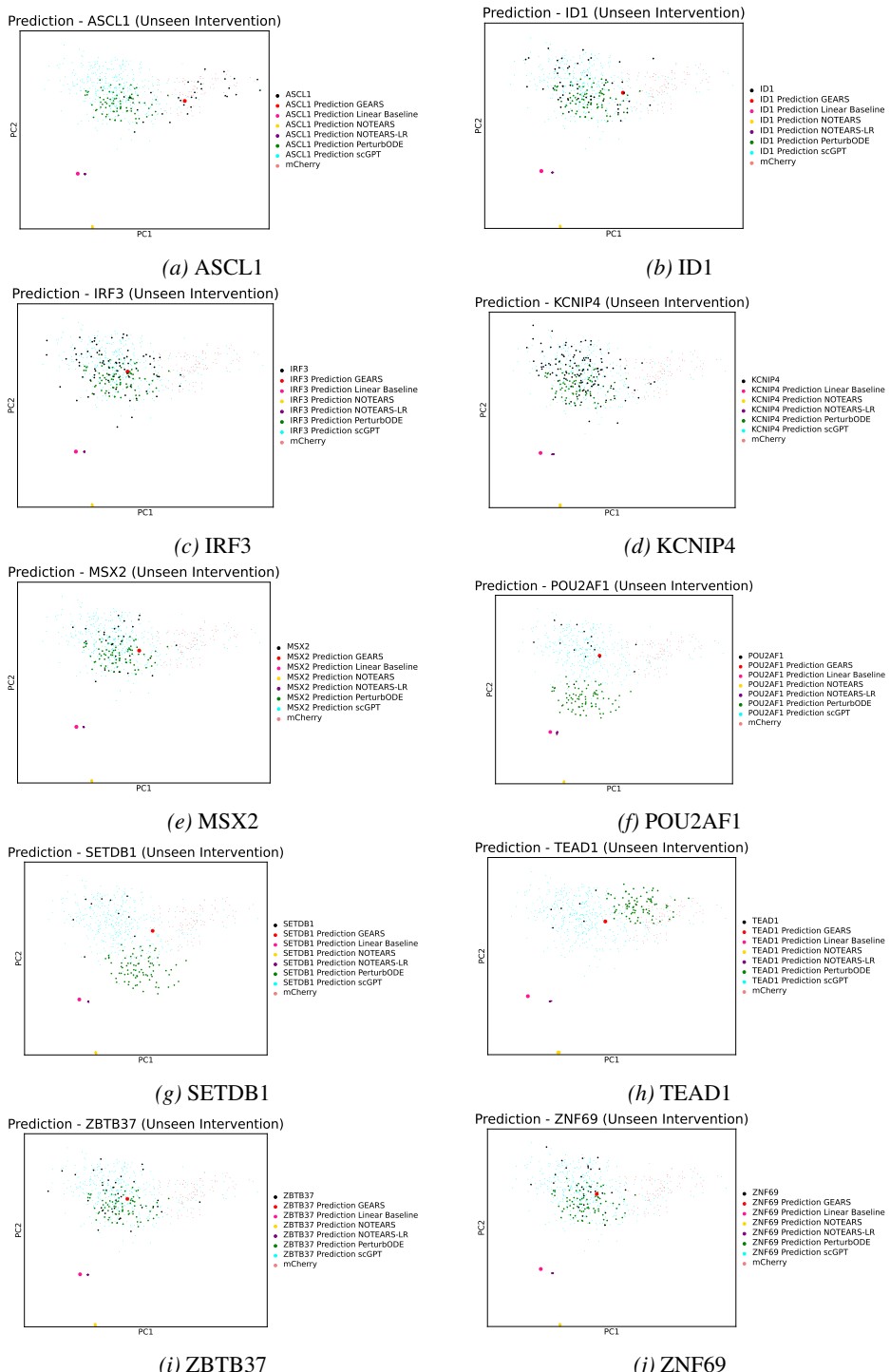

*Figure 20.* PCA of predictions on unseen interventions across models.

## A.18. Prediction on Seen Intervention UMAP and PCA Plots

PerturbODE: Ground Truth (Seen Intervention)

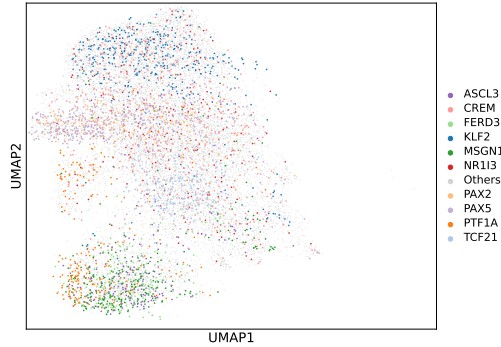

*(a)* Ground-truth embeddings

PerturbODE: Predictions (Seen Intervention)

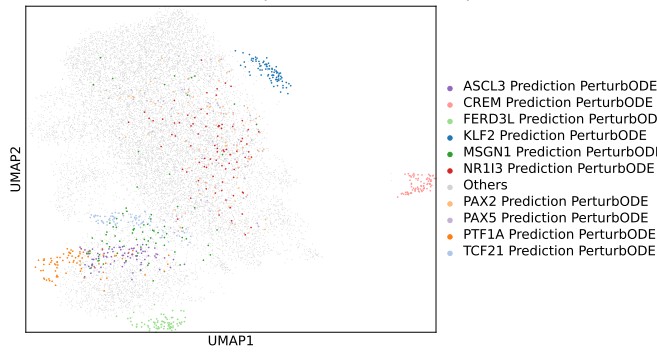

*(b)* Training-set predictions

*Figure 21.* UMAP projections of cell embeddings: 10 transcription factors (TFs) from the training set, the model's predictions on those TFs

## A.19. Ground truth GRNs from TF Atlas

The three GRNs with high confidence inferred in Joung et al. (2023) are consistent with their induced cell types and roles in development. GRHL1 and GRHL3 target TFAP2C and the TEAD family of TFs to induce trophoblasts, while FLI1 targets AP-1 family TFs (such as JUN and FOS) and ETV2 to induce vascular endothelial cells (Krendl et al., 2017; Dejana et al., 2007). The GRN consisting of CDX1, CDX2, and HOXD11 targeted posterior HOX genes is known to contribute to the definition of the anterior-posterior axis (Joung et al., 2023; Neijts et al., 2017). The three GRNs are in Figures 22, 23, 24. These GRNs are further supported by ATAC-seq peaks and motif analysis, as confirmed by Joung et al. (2023).

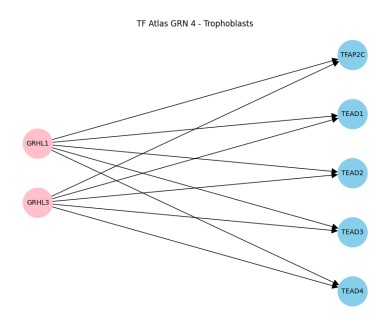

*Figure 22.* GRN with high confidence from TF Atlas - GRN8

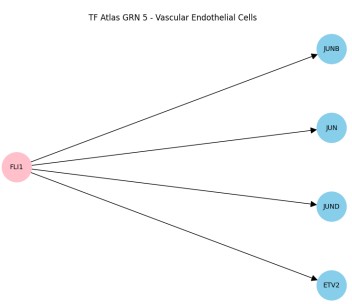

*Figure 23.* GRN with high confidence from TF Atlas - GRN4

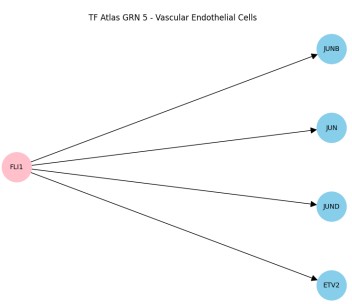

*Figure 24.* GRN with high confidence from TF Atlas - GRN5

## A.20. Inferred Modules Encapsulating Ground Truth GRNs and Gene Set Enrichment Analysis

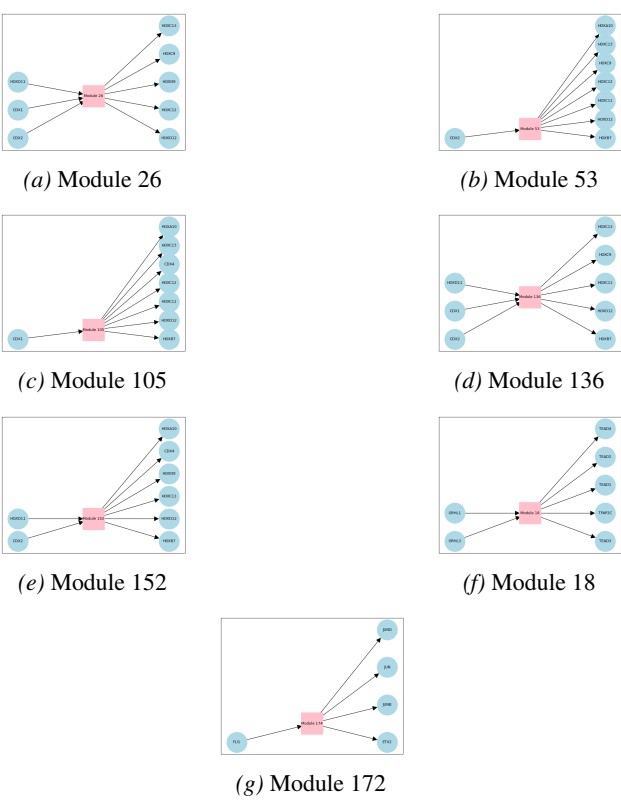

*(a)* Module 26          *(b)* Module 53

*(c)* Module 105          *(d)* Module 136

*(e)* Module 152          *(f)* Module 18

*(g)* Module 172

*Figure 25.* Modules identified by PerturbODE that align with established regulatory relationships.

### A.20.1. ANALYSIS OF INFERRED GENE MODULES

PerturbODE's framework enables direct interpretation of the inferred gene modules, which encapsulate multiple gene to gene interactions. These interactions are extracted from the $A$ and $B$ matrices (Equation 1), where the entries in $B$ represent directed edges from upstream genes to gene modules, and the entries in $A$ map the modules to downstream genes.

To highlight the advantages of PerturbODE's interpretability, we analyze the 200 inferred latent gene modules obtained from training on the TF Atlas dataset. We computed a test score based on the number of correct gene regulators and targets in the GRN selected by each module (Section A.4). We visualize seven modules with the highest scores in Figure 25, each corresponding to directed edges found in experimentally validated GRNs (Appendix A.19). The modules in (a) - (e) encapsulate the GRN responsible for specification of the anterior-posterior axis in development (Neijts et al., 2017). (f) and (g) successfully capture known GRNs responsible for inducing trophoblasts and vascular endothelial cells respectively (Krendl et al., 2017; Dejana et al., 2007).

Additionally, we compared the inferred modules to Erdős-Rényi random matrices in terms of the number of correct regulators and targets selected, yielding $p$-values of less than $0.001$ (Appendix A.4). Significant $p$-values indicate that the correct genes are not assigned to the modules by random chance. By inspecting the modules, we demonstrate that PerturbODE recovers the appropriate gene network structure, clustering genes from the same GRN and accurately inferring edges between them.

Gene Enrichment Analysis of Selected Modules - Clustered Heatmap

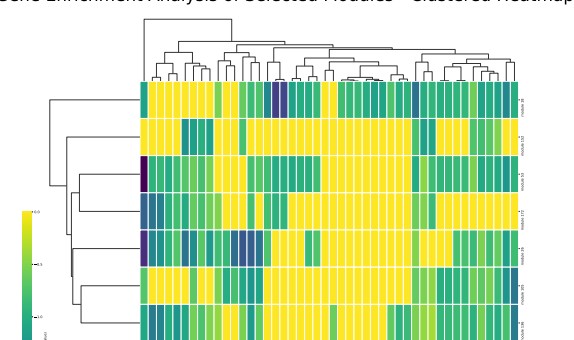

*Figure 26.* Gene enrichment clustered heatmap (average linkage) for selected modules.

We further validate PerturbODE's inferred gene modules through gene set enrichment analysis (GSEA), which evaluates the overlap between genes associated with known biological pathways and genes within each predicted module. Figure 26 presents a clustered heatmap of statistically significant pathway enrichments across modules (a) to (g), with details provided in Appendix A.20.2. Our analysis reveals biologically coherent patterns that align with cellular differentiation. Modules 172 and 136 show enrichment in pathways specific to vascular endothelial cells. Meanwhile, modules 26, 172, 136, 18, and 53 demonstrate strong enrichment in anterior-posterior (A-P) axis specification, with module 53 showing the strongest significance. Additionally, module 18 exhibits significant enrichment in pathways related to angiogenesis and fluid stress response.

### A.20.2. FULL GENE ENRICHMENT ANALYSIS

We performed gene enrichment analysis using the Reactome Pathway Database (2022) and the Gene Ontology Biological Process (2021) with hypergeometric test. The examined pathways were filtered to those relevant to the anterior-posterior axis and vascular endothelial cells. The upstream genes and downstream genes of each module are selected by taking those edges whose weights are greater

than 2 standard deviations of $B$ and $A$ respectively. Figure 29 illustrates the clustering of modules based on specific functions. A large number of modules exhibit enrichment for anterior-posterior specification— a pathway crucial in development. This observation is expected, considering that the TF Atlas comprises human embryonic stem cells.

To show that the modules are not selecting identical genes, we plotted histograms of genes selected by various modules. Figure 27 shows a histogram of genes selected by the highlighted modules we selected for evaluation in Section A.20.1, and Figure 28 showcases that of 10 randomly selected modules. Both histograms show clear clustering of gene selections by modules.

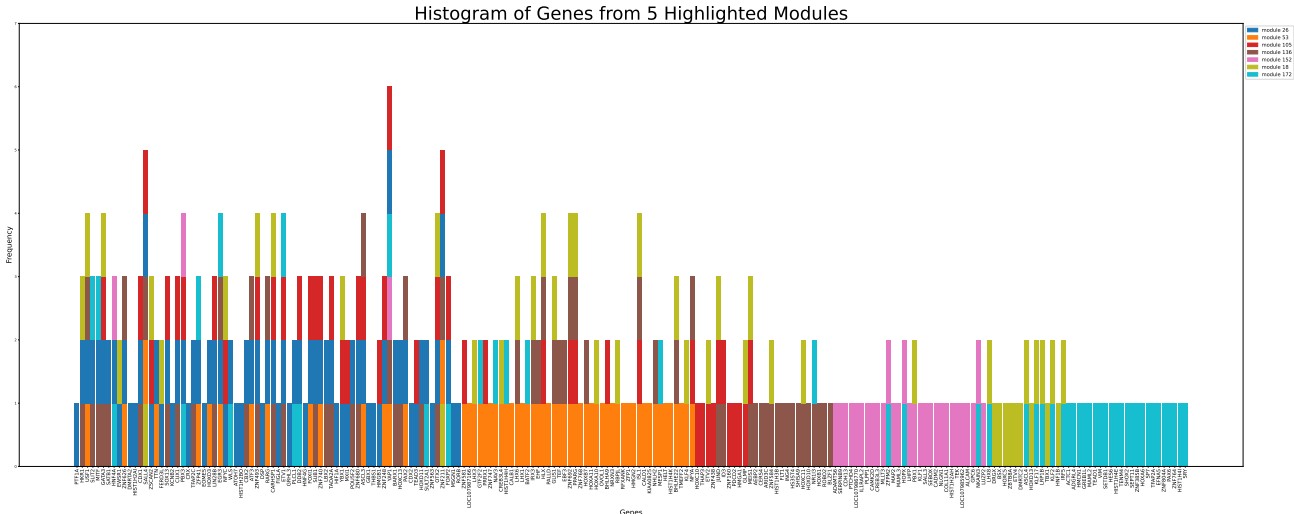

*Figure 27.* Histogram of Genes from 5 Highlighted Modules.

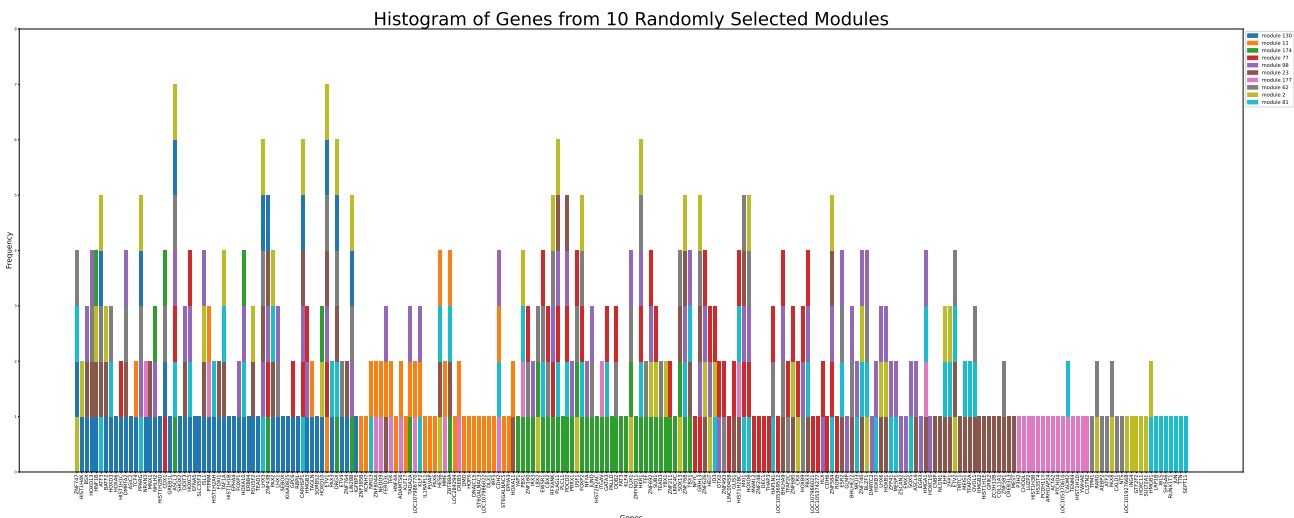

*Figure 28.* Histogram of Genes from 10 Randomly Selected Modules.

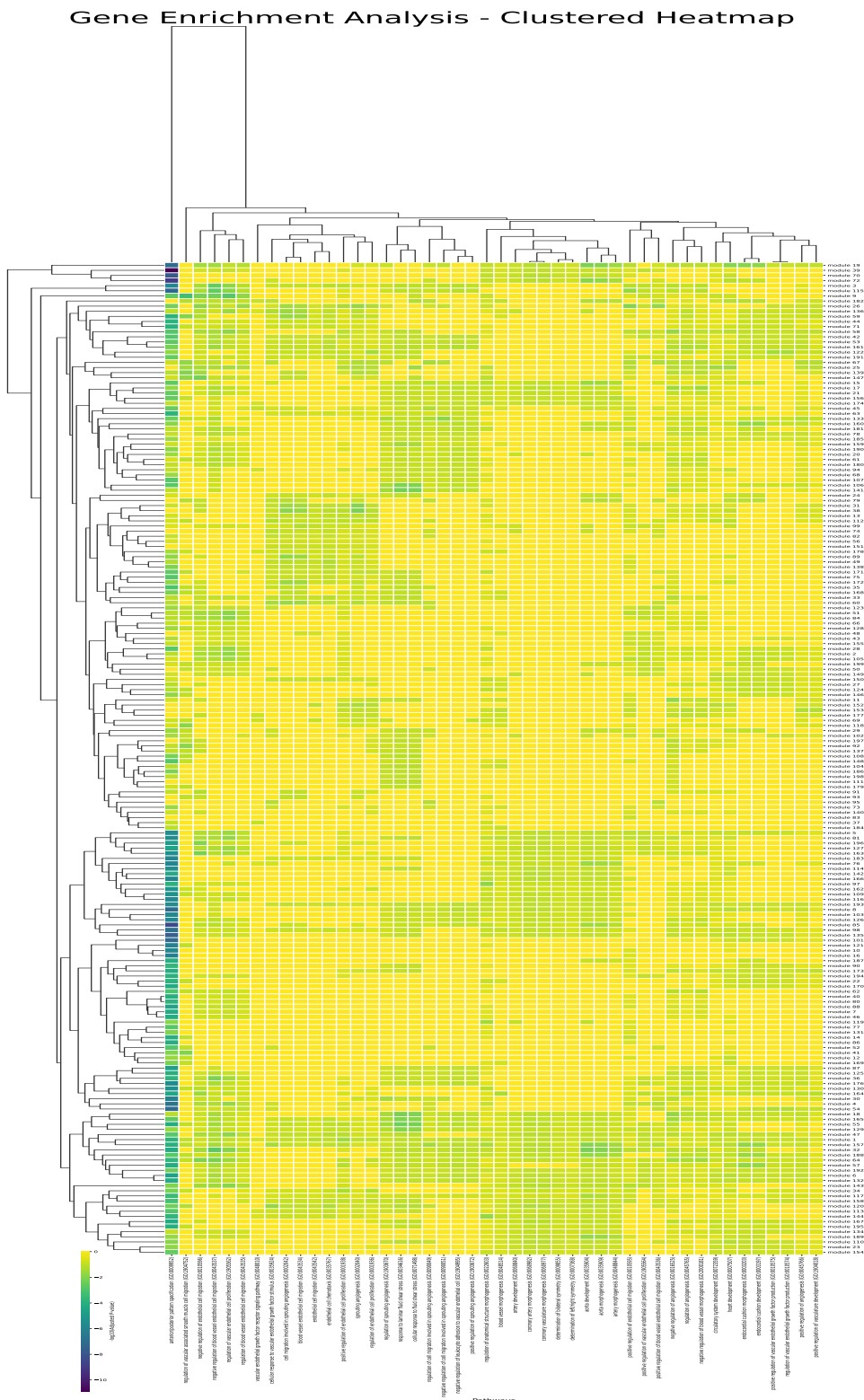

*Figure 29.* Gene enrichment clustered heatmap (average linkage)
for all modules.

## A.21. SERGIO simulation

SERGIO proposes simulation of scRNA-seq data by sampling a directed acyclic GRN through a SDE (Dibaeinia & Sinha, 2020). Although SERGIO does not support interventional data, we modified its framework to simulate gene overexpression with perfect interventions (CRISPR-a). For each interventional regime $I \in \mathcal{I}$, the SDE is parameterized in the following Equation 8.

$$dX_t = \left( M\big(P(X_t) - \lambda \circ X_t\big) + \sum_{j \in I} \gamma_j \cdot \delta_j \right) dt \\ + q \circ \left( \sqrt{P(X_t)}\, dW_\alpha + \sqrt{\lambda X_t}\, dW_\beta \right) \quad (8)$$

The infinitesimal change of expression level (which is the stochastic process $X_t$) of gene $j$ at time $t$ over an infinitesimal time interval $dt$, denoted as $(dX_t)_j$, is governed by its production rate $P_j(X_t)$, which is modulated by its regulators according to a given GRN in Equation 9. It also depends on the decay rate $\lambda \in \mathbb{R}^d_+$ and the noise amplitude $q \in \mathbb{R}^d$ influencing its transcriptional variability. $M$ and $\sum_{j \in I} \gamma_j \cdot \delta_j$ are the masking matrix and the overexpression term analogous to those in Equations 1 and 2.

$$P_j(X) = \sum_{j=0}^{d} p_{ji}(X) + b_j \quad \text{for } p_{ji} \text{ in } 10, 11 \quad (9)$$

$$p_{ji}(X) = K_{ji} \frac{X_i}{h + X_i}, \quad (10)$$
if regulator $i$ is an activator of gene $j$.

$$p_{ji}(X) = K_{ji} \left( 1 - \frac{X_i}{h + X_i} \right), \quad (11)$$
if regulator $i$ is a repressor of gene $j$.

For each pair of genes $i$ and $j$, the coefficients are initialized as in 12.

$$\begin{aligned}
\lambda_j &\sim \mathcal{N}(0.8, 0.2)_+, \\
K_{ji} &\sim \mathcal{U}(0, 5), \\
q_j &\sim \mathcal{U}(0.3, 1), \\
\gamma_j &\sim \mathcal{N}(10, 1)_+, \\
h &= \frac{1}{d} \sum_{j=1}^{d} \frac{b_j}{q_j}, \\
b_j &\sim \mathcal{N}(10, 0.01)_+ \quad \text{if gene } j \text{ is a master regulator,} \\
b_j &= 0 \quad \text{if gene } j \text{ is not a master regulator.}
\end{aligned} \quad (12)$$

$W_\alpha, W_\beta \in \mathbb{R}^d$ are two independent Wiener processes. We numerically simulate the SDE in Equation 8 using the Euler-Maruyama Scheme (E et al., 2019) with $\Delta t = 2$ in 50 steps.

$$(X_j)_{t+\Delta t} = (X_j)_t + \left( \left( P_j(X_t) - \lambda_j X_j(t) \right) \cdot \mathbb{I}_{j \notin I} + \gamma_j \cdot \mathbb{I}_{j \in I} \right) \Delta t \\ + q_j \sqrt{P_j(X_t)} \Delta W_\alpha + q_j \sqrt{\lambda_i X_j(t)} \Delta W_\beta \quad (13)$$

$$(\Delta W_\alpha)_j \sim \sqrt{\Delta t}\mathcal{N}(0, 1), \quad (\Delta W_\beta)_j \sim \sqrt{\Delta t}\mathcal{N}(0, 1) \quad (14)$$

Lastly, the SDE 8 is initialized at the expected fixed point $X_0$ (where the drift of the SDE vanishes) with overexpression but without masking (perfect intervention). SERGIO assumes Jansen's Equality $E[p_{ji}(X_i)] \approx p_{ji}(E[X_i])$ for simplicity of initialization (Dibaeinia & Sinha, 2020). Hence, $X_0$ is initialized to the following expectations in Equations 15 and 16:

$$E[X_j] = \frac{\sum_{i=0}^{d} p_{ji}(E[X_i])}{\lambda_j} + \gamma_j \cdot \mathbb{I}_{j \in I} \quad (15)$$
if $j$ is not a master regulator

$$E[X_j] = \frac{b_i}{\lambda_j} + \gamma_j \cdot \mathbb{I}_{j \in I} \quad \text{if gene } j \text{ is a master regulator} \quad (16)$$

When simulating data using SERGIO, we use a real yeast GRN ($dim = 400$) and 10 random DAGs ($dim = 100$) with 500 binary entries (1 or 0). For clarity of comparison across models, the real yeast GRN is pruned to enforce acyclicity and include only activator edges. For both scenarios, the synthetic dataset generated by SERGIO includes 10,100 cells, created from 100 intervention schemes, each targeting 5 genes, along with one non-intervention scheme. Each regime provides 100 observations.

## A.22. BEELINE (BoolODE) simulation

BEELINE (BoolODE) (Pratapa et al., 2020) is a single-cell RNA-seq simulator that models mRNA dynamics using stochastic differential equations (SDEs), where gene regulatory interactions are represented through Hill functions approximating Boolean logic. Here, $x_i$ denotes the mRNA concentration of gene $i$, $p_i$ represents its corresponding protein level, and $R_i$ reflects the concentration of regulatory proteins controlling gene $i$. The coefficients $m$, $r$, $l_p$, $l_x$,

and $s$ are constants, while $f(\cdot)$ captures the regulatory relationships within the GRN. The SDE system is defined as:

$$\frac{d[x_i]}{dt} = mf(R_i) - l_x[x_i] + s\sqrt{[x_i]}dB_t \qquad (17)$$

$$\frac{d[p_i]}{dt} = r[x_i] - l_p[p_i] + s\sqrt{[p_i]}dB_t \qquad (18)$$

Following the approach in SERGIO, we simulate perturbations by performing perfect interventions combined with overexpression (CRISPR-a). This involves suppressing the regulatory influence of the target gene in $f(\cdot)$ while introducing an additive shift to model overexpression. The overexpression is set to 20. This generates a collection of distributions in gene expression space, with each distribution corresponding to a distinct interventional condition. The environments include an observational (no-intervention) setting and multiple interventional settings where each gene is individually targeted. The resulting empirical distributions are denoted as $\{\rho_i\}_{i=0}^{k}$. The coefficients are set to the default given by the authors, where $r = 10$, $s = 10$, $l_x = 10$, $l_p = 10$, and $m = 20$.

We utilized the toy-GRN with bifurcating convergent behavior manually curated by the authors. We simulated 300 cells for each perturbation, and the simulation time is set to 1000 to encourage steady-state distributions of cells. There are 10 genes in this prescribed system, and we simulated perturbations with a single gene at a time for all 10 genes. We also simulated an observational distribution without perturbations.

### A.23. Gene Module Example: Flagella of E. coli

It is well established that the regulatory circuit responsible for flagellar production in E. coli follows the network motif of multiple-output Feedforward Loop (Alon, 2006, pp. 64-68). Its circuit is shown on the left of Figure 30, where FlhDC and FliA regulate $Z_1$, $Z_2$, and $Z_3$, which are operons encoding the proteins that make up the flagella of E. coli. (In fact, there are in total 6 operons for this process.) Each operon consists of a group of genes, and it is regulated by a weighted sum of non-linearly activated signals from FlhDC and FliA through Hill functions.

The order in which the operons are activated matches the order of proteins needed to assemble the flagella. The timing of activation is achieved by different activation thresholds in the Hill functions. If $Z_1$ is activated before $Z_2$, which is activated before $Z_3$, then $K_2 < K_3 < K_4$. In other words, $Z_1$ needs a lower concentration of FliA to be switched on. For example, $Z_1$ would include the group of genes encoding the proteins

for the MS ring (base of flagella) and $Z_3$ would be for the filament (tail of flagella). In PerturbODE, the activation threshold is tuned by the bias term, $\beta$, to the hidden neurons.

This structure can be represented in a two-layer MLP shown on the right of Figure 30. Each operon $Z_i$ is regulated by the weighted sum of signals from two modules $M_i$ and $M_i'$. The signals from FliA and FlhDC are first activated by Hill functions with different activation thresholds before being transferred to modules $M_i$ and $M_i'$ respectively.

To represent this gene regulatory circuit with an adjacency matrix $\mathbf{G} = A\,\mathrm{diag}(\alpha)B$, we multiply the two coefficient weight matrices of the MLP with an additional scaling to account for the rate of activation controlled by $\alpha$.

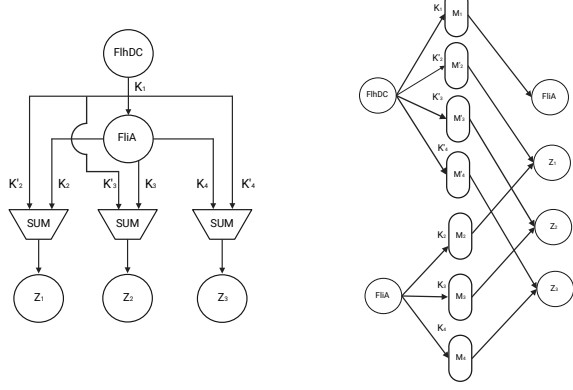

*Figure 30.* Regulatory circuit for the production of flagella in E. coli.

## A.24. Identifiability Proof

**Theorem A.1** (Identifiability of the shared drift parameterization). *Let $\Omega \subseteq \mathbb{R}^d$ be open and connected, and consider the shared drift*

$$v(x) = A\sigma(Bx) - Dx,$$

*where $\sigma : \mathbb{R}^m \to \mathbb{R}^m$ is an elementwise twice continuously differentiable activation function. For each constant intervention $c$, let $p_c$ be a positive continuously differentiable stationary density on $\Omega$ satisfying*

$$0 = -\nabla \cdot \big(p_c(v+c)\big).$$

*Suppose that the same family of stationary densities is also generated by an alternative drift*

$$\hat{v}(x) = \hat{A}\sigma(\hat{B}x) - \hat{D}x.$$

*Assume that there exists a nonempty open set $U \subseteq \Omega$ such that, for every $x \in U$, the score-difference vectors*

$$\{\nabla \log p_{c_i}(x) - \nabla \log p_0(x)\}_{i=1}^n$$

*span $\mathbb{R}^d$. Assume further that:*

1. *the rows of $B$ and $\hat{B}$ are nonzero and have unit norm;*

2. *the matrices $A$, $\hat{A}$, $B^\top$, and $\hat{B}^\top$ have full column rank;*

3. *$\sigma''$ vanishes only on a set of measure zero;*

*Then the parameters are identifiable up to permutation and sign symmetries of the hidden units. More precisely, there exist a permutation matrix $P$ and a diagonal matrix $\Lambda$, with diagonal entries in $\{-1, 1\}$, such that*

$$\hat{B} = \Lambda P^\top B, \qquad \hat{A} = A P \Lambda, \qquad \hat{D} = D.$$

*Equivalently, after aligning the hidden units of the two parameterizations, one has*

$$A = \hat{A}, \qquad B = \hat{B}, \qquad D = \hat{D},$$

> **Notation**
>
> *We write $\sigma : \mathbb{R}^m \to \mathbb{R}^m$ for an elementwise activation function. Its Jacobian evaluated at $z \in \mathbb{R}^m$ is denoted by $\sigma'(z) \in \mathbb{R}^{m \times m}$. Since $\sigma$ acts elementwise, $\sigma'(z)$ is diagonal. For a diagonal matrix $M \in \mathbb{R}^{m \times m}$, we write $\mathrm{mdiag}(M) \in \mathbb{R}^m$ for the vector containing the diagonal entries of $M$. Furthermore, $\sigma''(z)$ denotes the Jacobian of $\mathrm{mdiag}(\sigma'(z))$, which is also diagonal because $\sigma$ acts elementwise.*

*Proof.* Dividing $0 = -\nabla \cdot \big(p_c(v+c)\big)$ by $p_c(x) > 0$ gives

$$0 = \nabla \log p_c^\top (v+c) + \nabla \cdot v. \tag{19}$$

Let $c_0$ denote the control condition. Subtracting the control equation from the equation associated with intervention $c_i$ yields

$$\big(\nabla \log p_{c_i} - \nabla \log p_0\big)^\top v = -\nabla \log p_{c_i}^\top c_i. \tag{20}$$

By assumption, the same equation applies to $\hat{v}$, so taking the difference implies

$$\big(\nabla \log p_{c_i} - \nabla \log p_0\big)^\top (v - \hat{v}) = 0. \tag{21}$$

We have assumed that there exists an open set $U$ such that, for every $x \in U$, the vectors

$$\{\nabla \log p_{c_i}(x) - \nabla \log p_0(x)\}_{i=1}^n$$

are linearly independent. It then follows that, for every $x \in U$,

$$A\sigma(Bx) - Dx = \hat{A}\sigma(\hat{B}x) - \hat{D}x \tag{22}$$

Rearranging (22), and using the vectorization identity of the column-wise Khatri rao product $\odot$:

$$\begin{aligned} vec(\hat{D} - D) = &(\hat{B}^T \odot \hat{A})mdiag(\sigma'(\hat{B}x)) \\ &- (B^T \odot A)mdiag(\sigma'(Bx)) \end{aligned} \tag{23}$$

So another Jacobian gives

$$(B^T \odot A)\sigma''(Bx)B = (\hat{B}^T \odot \hat{A})\sigma''(\hat{B}x)\hat{B} \tag{24}$$

We next identify the individual rows of $B$. Let $b_i^\top$ denote the $i$-th row of $B$. Choose $x, x + \epsilon \in U$ such that

$$\epsilon \perp b_i \iff i \neq 1.$$

Evaluating (24) at $x$ and $x + \epsilon$ and subtracting gives

$$\begin{aligned} (B^\top \odot A) \left(\sigma''(B(x + \epsilon)) - \sigma''(Bx)\right) B \\ = (\hat{B}^\top \odot \hat{A}) \left(\sigma''(\hat{B}(x + \epsilon)) - \sigma''(\hat{B}x)\right) \hat{B}. \end{aligned} \tag{25}$$

By construction, all diagonal entries of $\sigma''(B(x + \epsilon)) - \sigma''(Bx)$ vanish except the first. Hence, the left-hand side of (25) is a nonzero rank-one matrix.

$\epsilon$ is orthogonal to all but one row of $\hat{B}$. Without loss of generality up to permutation, we assume that it is also $\hat{b}_1$. Consequently, both sides of (25) have row spaces spanned by $b_1^\top$ and $\hat{b}_1^\top$, respectively, and therefore

$$\text{span}(b_1) = \text{span}(\hat{b}_1).$$

If the rows of $B$ and $\hat{B}$ are normalized to unit norm, then

$$b_1 = \pm \hat{b}_1.$$

Repeating the argument for each row and accounting for the common permutation of hidden units yields

$$B = P\Lambda\hat{B},$$

where $P$ is a permutation matrix and $\Lambda$ is a diagonal matrix with diagonal entries in $\{-1, 1\}$.

Having identified $B$ up to permutation and sign symmetries, we align the two parameterizations so that $B = \hat{B}$. Then Equation (24) implies

$$B^\top \odot (A - \hat{A}) = 0,$$

Since every row $b_i^\top$ of $B$ is nonzero, each column satisfies

$$b_i \otimes (a_i - \hat{a}_i) = 0,$$

and hence $A = \hat{A}$. Finally, substituting $A = \hat{A}$ and $B = \hat{B}$ into Equation (22) yields $Dx = \hat{D}x$ for all $x \in U$. Since $U$ is open, it follows that $D = \hat{D}$.

$\square$

## A.25. Computational Resources

All methods except GENIE3 were run on GPU nodes with V100 GPU cards (32 GB memory), with a maximum runtime of 24 hours per experiment. GENIE3 was executed on a CPU with 50 GB of memory and a runtime limit of 72 hours. The storage required for the TF Atlas dataset is 4.1 GB.

## A.26. Code Availability

Code for reproducing the experiments will be available at `https://github.com/daklab/PerturbODE`.

