# OpenReview forum: "Interpretable Neural ODEs for Gene Regulatory Network Discovery under Perturbations"
_ICML.cc/2026/Conference — ICML 2026 regular_

### Official Review · Reviewer_7vZh · 2026-03-11

**Soundness:** 3
**Presentation:** 1
**Significance:** 2
**Originality:** 2
**Overall Recommendation:** 3
**Confidence:** 4

**Summary:**

The paper proposes to model cell state trajectories in biological datasets using ODE to potentially capture the non-linear dynamics of its evolution under perturbation. From the learned ODE parameters, they infer the underlying causal Gene Regulatory Network (GRN) and use the model to simulate the effects of unseen genetic perturbations.

**Compliance With Llm Reviewing Policy:**

Affirmed.

**Final Justification:**

I do appreciate the authors' further response. I still feel the work in its current version needs major revision for publication, particularly in terms of novelty and theoretical justification, as pointed out in my acknowledgment. The score will remain unchanged.

**Key Questions For Authors:**

1. Why FDR values are so large in Table 1? It seems suggesting the difficulty levels of the tasks are high.

2. How stable is training the proposed method? Backpropagation-through-time in neural ODEs normally face challenges with stability.

3. How reasonable is it to assume that the interactions represented by A and B do not change over time?

**Limitations:**

No limitations are discussed.

**Strengths And Weaknesses:**

# Strengths
- The paper is clearly written.

# Weaknesses
- Technical novelty seems incremental. The idea of applying neural ODE for perturbation effect prediction is reviewed even in the related work section. It's unclear to me what is the technical novelty of the proposed method. Specifically, it's not clear the difference between the proposed approach from existing ODE-based methods discussed in Section 2. It'll be helpful to explicitly differentiate the ODE in Equation (1) from those in Jackson et al. and the linear drift term in Bicycle (Rohbeck et al.), and clarify the advantages of the proposed functional form.

- The performance improvements over the baseline methods, such as GENIE3, are not consistently significant across test datasets.
- It's unclear how the settings of the neural network, such as architecture (2 layers?), learning rates, and hyper-parameter setting, is determined. Some ablation study would be helpful.

- The GRN is estimated as G = A diag($\alpha$)B, using the parameters of the ODE, but its biological justification is not clearly explained.

- Regarding the GRN formula, it also seems the GRN is not uniquely associated with a particular dynamics. A, B and $\alpha$ can be modified in ways that can result in the same GRN, but has different dynamics (defined in equation 1).

- The performance improvement in Figures 2, 3 and Tables 1, 2b are not significant. In some cases, existing baselines outperform PerturbODE.

- The evaluation metrics are inconsistently used. Besides AUPRC, sometimes only recall is reported.

- The presentation is poor. Fontsizes across figures are inconsistent.

---

> ### Author Rebuttal · Authors · 2026-03-30
>
> ***W1: Differentiate the ODE from Jackson and Rohbeck***
>
> - [2] relies on preprocessing pseudotime, which PerturbODE does not. Removing pseudotime makes the framework more robust. The BEELINE GRN review paper concludes poor performance of pseudotime-based methods.
> - All the measured cells fit along a single pseudotime trajectory in [2], while our method is capable of mapping the dynamics of entire cell populations.
> - [2] does not encode GRN explicitly. To extract the GRN, they compute coefficients of partial determination, which is less interpretable.
> - Compared to [1] and [2], PerturbODE has a gene module (biological pathways) mechanism. Further, this reduces the computational and statistical complexity from learning n^2 variables to 2n x d (n = number of genes, d =  number of modules).
>
> ***W2: Performance not significant or consistent.***
>
> See response to weakness 1 from reviewer SvWn.
>
> ***W3: Ablation for (2 layers?), learning rates, and hyper-parameter***
>
> - A 2-layer MLP is chosen to make the model interpretable. If we have more than 2 layers, we need to use methods st saliency map, which doesn't work as well. See below (benchmark on all 223 genes in ChiP-Atlas):
>
>   | Model (5 runs)                        | (AUPRC) Mean   | Std   |
>   |--------------------------------------|-------:|------:|
>   | 3-layer MLP + saliency GRN inference | 0.0151 | 0.001 |
>   | 2-layer MLP (standard)               | 0.022  | 0.004 |
>
> - We reported reasons for hyperparameter choices in Appendix A.2.1, including **number of hidden neurons and numerical methods** of the ODE solver.
>
> - We report the ablation study of **learn-rate**. The inference performance is consistent when there isn't OOM errors.
>
>   | learn-rate | AUPRC |
>   |---------------|------:|
>   | 1             | OOM |
>   | 0.1           | OOM |
>   | 0.01          | 0.021 |
>   | 0.001         | 0.021 |
>   | 0.0001        | 0.022 |
>
> - We included the **AUPRC v.s. number of modules** in response to reviewer SvWn. Also **validation loss v.s. number of modules** is in A.2.1 Figure 9.
>
> - We also included ablation study in Figure 4 with **different versions of perturbation encodings**.
>   - PerturbODE*: imperfect intervention + learnable over-expression strength
>   - PerturbODE-I: imperfect intervention
>   - PerturbODE-P: perfect intervention
>
> - Furthermore, we included **power and ablation study** in section A.9, where as we access model by increasing the number of perturbations and changing L1 penalty values.
>
> ***W4: G = A diag(a)B, biological justification?***
>
> Our GRN is decomposed into gene modules. In complex cells, sets of functionally related genes have highly correlated expression patterns across cell-types and conditions. The **“gene modules” (pathways)** are typically co-regulated by specific transcription factors. The inferred gene modules can be evaluated via gene set enrichment analysis in Appendix A.12.1 and in Figure 27.
>
> Further, in Appendix A.15, we worked through **a real example in E-coli** where the regulatory circuit is well represented by our GRN formulation.
>
> ***W5: GRN identifiability***
>
> Under the assumptions of stationary equilibrium of the observed perturbed distributions, lack of self-cycle, and disentanglement of gene modules, we provided an identifiability proof in A.16.  We acknowledge that identifiability requires some mathematical assumptions on the underlying GRN, however this is a common constraint on GRN inference methods (for example, see the proof of identifiability for DCDI in [3]).
>
> ***W6: Besides AUPRC, recall is reported.***
>
> For TF Atlas, in Fig 6., we also used AUPRC to benchmark our inferred GRN against reference GRN with both positive and negative edges. Hence, we disagree that "sometimes only recall is reported".
>
> We included recall to validate against high-confidence reference GRN edges to be more rigorous. We only reported recall because these high-confidence edges include only positive edges, giving us only true positives and false negatives.
>
> ***Q1: FDR***
>
> FDR is high due to the high difficulty level of the problem, and we outperform other methods in terms of FDR at most sparsity levels.
>
> ***Q2: Stability***
>
> Stability + generalizability analysis in A.17, via bootstraps and alternating train-val splits find true GRN edges to be selected 50-70% of the time, but even with slightly wider error bars we still outperform competing methods as in section 4.2
>
> ***Q3: GRN time-invariant?***
>
> In terms of biophysics, cells don't have knowledge about time. Rather the expression changes depend on the state space of transcriptional landscape. Some methods introduce time into GRN to account for unobserved variables. But, since PerturbODE scales to thousands of genes, it is reasonable to assume time-invariant GRN.
>
>
> [1] Rohbeck 2024.
>
> [2] Jackson, C. 2024.
>
> [3] Brouillard, Philippe, et al. "Differentiable causal discovery from interventional data." Advances in Neural Information Processing Systems 33 (2020): 21865-21877.

---

> > ### Author Rebuttal · Reviewer_7vZh · 2026-04-04
> >
> > I appreciate the authors' rebuttal.
> >
> > My concern on identifiability is somewhat addressed. For W4, the response is not quite clear to me. My concern on the consistency of performance improvement is not convincing to me either.

---

> > > ### Author Response · Authors · 2026-04-04
> > >
> > > We appreciate the reviewer’s additional comments.  We elaborate more on the raised issues below:
> > >
> > > Regarding modeling of the GRN, we model the interaction between genes in the ODE via $A  \\sigma\\left( \\alpha \\circ (B y^{(r)}(t) - \\beta) \\right)$, where $y^{(r)}(t)$ is the gene expression vector at time $t$.  Here, $B$ maps from the space of all genes to the smaller space of gene modules, and $A$ maps back.  There are many compatible biological interpretations of gene modules, either as formations of protein complexes from various transcription factors or as sets of functionally related genes.
> > >
> > > The additional parameters $\\alpha$ and $\\beta$ act as scaling and bias terms in a two-layer MLP, and $\\sigma$ is a non-linear activation function equivalent to the Hill function, which is commonly used to model promoter activity (see Appendix A of Alon 2006 [1]).
> > >
> > > Since $\\sigma$ is monotonically increasing, and $\\alpha$ and $\\beta$ are strictly positive, we extract the GRN by taking a Taylor expansion of the drift and approximating the Jacobian of $\\sigma$ as the identity, computing $G = A diag(\\alpha) B$.
> > >
> > > As an example, suppose gene X and gene Y are regulators of gene Z via a single gene module $M_1$, then the GRN-driven transcription rate of Z can be written as,
> > >
> > > $$T_z = a_1 \\sigma(\\alpha_1 \\cdot (b_1 X + b_2 Y - \\beta_1 ))$$
> > >
> > >
> > >
> > > Once the value of the linear combination, $b_1 X + b_2 Y$, passes a module-specific activation threshold set by $\\beta_1$, the combined signal is non-linearly activated by the sigmoid function, going quickly from 0 to 1 (see 1.3.4 of Alon 2006 [1]). The module-specific activation speed is further modulated by $\\alpha_1$. Lastly, $a_1$ determines the strength and sign of $M_1$’s contribution to gene Z.
> > >
> > > One can find similar real examples of this recurring network motif in the multiple-output feedforward loops in E. coli's regulatory circuits (see 4.4 of Alon 2006 [1] or Appendix A.15 of our submission).
> > >
> > > Regarding the consistency of performance improvement, our key observation is that on real-world data from the TF Atlas (Figure 4, 5, and 6), PerturbODE achieves the largest relative gains compared to baseline methods. This is because the higher-dimensional data is very hard to learn from without dimension reduction, which our method does implicitly through the projection to the smaller number of gene modules. This fact has been observed in other works on gene network recovery [2].
> > >
> > > [1] Alon, U. An Introduction to Systems Biology: Design Principles of Biological Circuits. CRC Press Taylor & Francis Group, A Chapman & Hall Book, 2006.
> > >
> > > [2] Segal, E., Shapira, M., Regev, A., Pe’er, D., Botstein, D., Koller, D., & Friedman, N. (2003).
> > > Module networks: Identifying regulatory modules and their condition-specific regulators from gene expression data. Nature Genetics, 34(2), 166–176. https://doi.org/10.1038/ng1165

---

### Official Review · Reviewer_Ty9U · 2026-03-13

**Soundness:** 4
**Presentation:** 4
**Significance:** 4
**Originality:** 4
**Overall Recommendation:** 5
**Confidence:** 4

**Summary:**

This paper proposes PerturbODE a new model based on Neural ODEs which models cell trajectories under perturbations originating from a control unperturbed cell and moving toward the target perturbed population. One key novelty is that ODE is formulated such that it explicitly parameterizes a gene regulatory network (GRN, usually a causal DAG defined over gene nodes) through a single hidden layer MLP. This MLP decomposes GRN into two interpretable low rank matrices whose parameters are learned from perturbational single-cell RNAseq datasets. Therefore, this provides a tool for inferring GRN from single-cell transcriptome data. Additionally, the ODE formulation explicitly incorporates gene knockdown or overexpressions which provide the opportunity for in-silico evaluations of perturbation effects after model training. Authors have evaluated the performance of their model in inferring underlying GRN using both simulated and real data. For benchmarking on real data, since the ground truth GRN that underlies human biology is not yet reliably resolved and known, authors have benchmarked their models against GRNs obtained from various data sources including RNA-seq, ATAC-seq, and GRN edges previously validated by CRISPR experiments.

**Compliance With Llm Reviewing Policy:**

Affirmed.

**Key Questions For Authors:**

- It’s interesting (yet not uncommon) that the authors chose to minimize a distribution-level metric, Wasserstein-2 distance (WD), as the loss function, which penalizes differences between the population/distribution of true and predicted perturbed cells. An ablation by replacing or augmenting this loss with MSE (given properly normalized data), or negative binomial reconstruction loss (given raw counts) on pseudo-bulked expressions, would be informative. Also, I understand the biological motivation behind the sparsity required for matrix B, but I was wondering whether the authors tested removing the L1 norm of B from the loss shown in Equation 5 (or replacing it with an L2 norm), and if so how this affected performance and interpretability.
- AUPRC is a good metric and generally one of the most sensitive and reflective metrics for GRN inference. Due to the class imbalance, it would be helpful to also report the random AUPRC baseline (i.e. the positive class rate).
- Perturbation response prediction was only evaluated using Wasserstein distance (WD) and cell-type distance, which may not be fully reflective of “perturbation response,” i.e. the change between control and perturbed populations. Delta-based metrics, which measure the agreement between true and predicted differences in pseudo-bulked expression, or DEG recovery metrics that measure the recall, precision, and/or F1 score of predicted DEGs versus true DEGs, are generally more reflective of response prediction. I recommend the authors add at least one such metric.

Minor comments:
- The term significant was used in a couple of sentences, for example “PerturbODE significantly outperforms all linear methods in terms of W2 distance and cell-type distance with held-out interventions (Table 2a and Table 2b)”. It would be better to reserve “significant” for cases supported by statistical testing.
- It seems that equation 1 could in principle allow for a more expressive GRN parameterization with deeper networks, for example by replacing matrix A with a nested logistic sigmoid-activated MLP or another architecture. I guess the main challenge would then be interpretation of this deeper module to recover learned gene-gene interactions, perhaps through attribution methods? I was wondering whether the authors have thought about this direction and what their perspective is.

**Limitations:**

Yes.

**Strengths And Weaknesses:**

The manuscript is well written and clearly communicates the modeling details, training pipeline, and evaluations. The evaluations also seem comprehensive, as they include both simulated and real datasets, multiple sources for compiling real GRNs, and comparisons against relevant competing models for each task. While one key application of the model is GRN inference, the follow-up experiments for zero-shot prediction of perturbation effects are nice and show competitive performance against some models designed specifically for perturbation response prediction. While the proposed model is not the top performer in all tasks, based on the reported results it consistently stays among the top models. I have some comments and concerns, but I believe the paper will be a valuable read for the ICML audience active in systems biology, so I assign the paper an overall score of 5.

---

> ### Author Rebuttal · Authors · 2026-03-30
>
> ***Question 1: Suggestions on MSE loss. Negative binomial loss on raw counts. Remove L1 penalty***
>
> Thank you for the suggestion of alternative distributional distances.  We now report the MSE between the mean of the predicted and true distributions (i.e. the L2 norm on the mean difference). We give this loss below (benchmarking on all 223 genes in the ChIP-Atlas):
>
> | Loss Function (5 runs)      (AUPRC)| Mean   | Std   |
> |-----------------------------|-------:|------:|
> | L2 (pseudobulk MSE)         | 0.0158 | 0.001 |
> | Wasserstein-2               | 0.022  | 0.004 |
>
>
> We also report ablation study by the removal of L1 penalty term below:
>
>
> | L1 Penalty (5 runs)     (AUPRC)| Mean   | Std   |
> |-----------------------------|-------:|------:|
> | Without L1 Penalty        |  0.017 | 0.001 |
> | With L1 Penalty               | 0.022  | 0.004 |
>
> Moreover, in appendix A.9, we conducted an ablation study of the model performance with **different L1 penalty**.
>
>
> Early on, we also attempted to train on **raw count space**, but, due to the noisy nature of single-cell sequencing data, especially in the presence of drop-outs, found worse performance. When the data is smoothened in the normalized and log-transformed space, we achieve much better performance in both GRN inference and perturbation prediction.
>
>
> ***Question 2: random AUPRC baseline***
>
> We report the AUPRC of expected random baseline below. PerturbODE consistently outperforms the random baselines.
>
> Figure 6: TF Atlas (ChiP-Atlas edges relevant to early embryonic development): 0.013
>
> Figure 13: TF Atlas (full ChiP-Atlas): 0.015
>
> Figure 2: BoolODE simulated data: 0.18
>
> Figure 3 (a): Sergio simulated data (random DAG 100 dimensions): 0.05
>
> Figure 3 (b): Sergio simulated data (yeast 400 dimensions): 0.0038
>
> ***Question 3: Quantify performance of perturbation predictions via DEGs***
>
> We thank the reviewer for their helpful suggestions. DEG makes lot of sense! For scGPT, NO-TEARS, NO-TEARS-LR, and PerturbODE, we computed DEGs using the Wilcoxon test. In contrast, GEARS and the Linear Baseline generate only a single predicted sample; therefore, DEGs were derived by ranking genes based on log fold changes against pseudobulked control cell population.
> To ensure a fair comparison with GEARS and the Linear Baseline, we additionally pseudobulked PerturbODE’s predictions and report the corresponding results in Rebuttal Table 2.
>
> **Rebuttal Table 1: F1 computed from top 50 DEG Wilcoxon (of the upregulated genes).**
> | perturbation |   NOTEARS | NOTEARS-LR | PerturbODE |     scGPT |
> | ------------ | --------: | ---------: | ---------: | --------: |
> | ASCL1        |      0.06 |       0.08 |       0.14 |      0.12 |
> | ID1          |      0.02 |       0.00 |       0.04 |      0.04 |
> | IRF3         |      0.02 |       0.00 |       0.04 |      0.02 |
> | KCNIP4       |      0.02 |       0.00 |       0.06 |      0.02 |
> | MSX2         |      0.02 |       0.00 |       0.06 |      0.04 |
> | POU2AF1      |      0.04 |       0.02 |       0.06 |      0.06 |
> | SETDB1       |      0.10 |       0.04 |       0.02 |      0.06 |
> | TEAD1        |      0.06 |       0.00 |       0.02 |      0.04 |
> | ZBTB37       |      0.02 |       0.04 |       0.12 |      0.06 |
> | ZNF69        |      0.06 |       0.00 |       0.06 |      0.06 |
> | **median**   |  **0.03** |   **0.00** |   **0.06** |  **0.05** |
> | **std**      | **0.026** |  **0.025** |  **0.037** | **0.029** |
>
>
> **Rebuttal Table 2: F1 computed from top 50 DEG obtained from delta gene rankings (upregulated genes).**
>
> | perturbation | GEARS | Linear Baseline | PerturbODE |
> |--------------|------:|----------------:|-----------:|
> | ASCL1        | 0.10  | 0.00            | 0.08       |
> | ID1          | 0.22  | 0.02            | 0.34       |
> | IRF3         | 0.36  | 0.02            | 0.34       |
> | KCNIP4       | NaN   | 0.04            | 0.38       |
> | MSX2         | 0.10  | 0.02            | 0.28       |
> | POU2AF1      | 0.26  | 0.00            | 0.30       |
> | SETDB1       | 0.04  | 0.04            | 0.20       |
> | TEAD1        | 0.24  | 0.08            | 0.30       |
> | ZBTB37       | 0.22  | 0.04            | 0.30       |
> | ZNF69        | 0.36  | 0.08            | 0.34       |
> | **median**   | **0.22** | **0.03**     | **0.30**   |
> | **std**      | **0.107** | **0.027**   | **0.091**  |
>
>
> ***Question 4: word choice of "significant"***
>
> We will remove the use of the word “significant”.
>
> ***Question 5: Use deeper networks to recover learned gene-gene interactions through attribution methods?***
>
> We empirically explored this architecture and attempted it with a 3-layer MLP, and we inferred the GRN via Saliency map. The results are worse than our standard approach. Comparison can be seen below:
>
> | Model (5 runs)                          (AUPRC)| Mean   | Std   |
> |----------------------------------------|-------:|------:|
> | 3-layer MLP + saliency GRN inference   | 0.0151 | 0.001 |
> | 2-layer MLP (standard)                 | 0.022  | 0.004 |

---

> > ### Author Rebuttal · Reviewer_Ty9U · 2026-04-03
> >
> > Thank you for the rebuttal. The authors have properly addressed my main comments and concerns. The additional ablation experiments help justify some of the model’s design choices, and the new DEG-based metric provides a more complete picture of performance on perturbation response prediction. I will maintain my positive score.

---

### Official Review · Reviewer_YWg7 · 2026-03-13

**Soundness:** 2
**Presentation:** 2
**Significance:** 2
**Originality:** 1
**Overall Recommendation:** 2
**Confidence:** 3

**Summary:**

Basically this paper proposes a new framework namely PerturbODE which uses neural ordinary differential equations to model the cell state trajectories under thousands of genetic perturbations.They embed the Gene Regulatory Network (GRN) directly into parameters of a 2-layer MLP (single-hidden-layer). Because of this setup, authors claim that model can do simultaneous trajectory inference, causal graph discovery using Wasserstein-2 optimal transport, and also it claims to predict unseen perturbations (?). Instead of just looking at individual genes, they group genes into modules which is making the whole system more interpretable biologically.

**Compliance With Llm Reviewing Policy:**

Affirmed.

**Final Justification:**

Methodological novelty of the paper is pretty limited. Using flow-matching and ODEs is already standard practice in single-cell modeling, making this approach feel more incremental than groundbreaking. Additionally, the evaluation feels incomplete. Synthetic datasets like SERGIO and BoolODE are fine for a sanity check, but they don't prove the model can handle the messy reality of actual perturbational scRNA-seq data. It needs benchmarking against established standards like DREAM Network Inference Challenge, BEELINE, CausalBench, and strong baselines including variants of PC, SCENIC and GRNBoost2. Therefore, my score is not going to change.

**Key Questions For Authors:**

Why does GEARS still outperform PerturbODE on W2?
Have you done wet-lab experiments to corroborate that your model generalizes well to unseen perturbations, cell lines, donors etc.?

**Limitations:**

yes

**Strengths And Weaknesses:**

Strengths:
1. Modeling the perturbation biology with nonlinear dynamics instead of just static graph estimation is a meaningful direction.
2. It is scalable if you compare it with classical directed acyclic graph (DAG) discovery methods.

Weaknesses:
1. GEARS is still doing better on the W2 metric in held out intervention comparison. PerturbODE mainly wins against the weaker or more limited baselines.
2. The model shows high variability and instability. Authors actually acknowledge this themselves, especially when it is tested on longitudinal time-series Perturb-seq data.

---

> ### Author Rebuttal · Authors · 2026-03-30
>
> ***Weakness 1: GEARS is still doing better on the W2 metric in held out intervention comparison. PerturbODE mainly wins against the weaker or more limited baselines.***
>
> PerturbODE is designed for GRN inference, and our main results lie in section 4.2, where we outperform all other methods in real biological data. Perturbation predictions are an extension task, where we show that our method outperforms simpler methods and is competitive with state of the art methods.
>
> Further, in biology, it is much more important to be able to predict the correct cell types or differentially expressed genes. Below, we report the F1 score of predicted DEGs versus true DEGs, where we outperform other methods.
> In Rebuttal Table 1, for scGPT, NO-TEARS, NO-TEARS-LR, and PerturbODE, we computed DEGs using the Wilcoxon test. In contrast, GEARS and the Linear Baseline generate only a single predicted sample; therefore, due to the lack of replicates, DEGs were derived by ranking genes based on log fold changes against pseudobulked control cell population.
> To ensure a fair comparison with GEARS and the Linear Baseline, we additionally pseudobulked PerturbODE’s predictions and report the corresponding results in Rebuttal Table 2. We also looked at upregulated genes since we work with a TF over-expression dataset.
>
>
> **Rebuttal Table 1: F1 computed from top 50 DEG Wilcoxon (of the upregulated genes).**
> | perturbation |   NOTEARS | NOTEARS-LR | PerturbODE |     scGPT |
> | ------------ | --------: | ---------: | ---------: | --------: |
> | ASCL1        |      0.06 |       0.08 |       0.14 |      0.12 |
> | ID1          |      0.02 |       0.00 |       0.04 |      0.04 |
> | IRF3         |      0.02 |       0.00 |       0.04 |      0.02 |
> | KCNIP4       |      0.02 |       0.00 |       0.06 |      0.02 |
> | MSX2         |      0.02 |       0.00 |       0.06 |      0.04 |
> | POU2AF1      |      0.04 |       0.02 |       0.06 |      0.06 |
> | SETDB1       |      0.10 |       0.04 |       0.02 |      0.06 |
> | TEAD1        |      0.06 |       0.00 |       0.02 |      0.04 |
> | ZBTB37       |      0.02 |       0.04 |       0.12 |      0.06 |
> | ZNF69        |      0.06 |       0.00 |       0.06 |      0.06 |
> | **median**   |  **0.03** |   **0.00** |   **0.06** |  **0.05** |
> | **std**      | **0.026** |  **0.025** |  **0.037** | **0.029** |
>
>
> **Rebuttal Table 2: F1 computed from top 50 DEG obtained from delta gene rankings (upregulated genes). Predictions from PerturbODE are pseudobulked to enable comparison.**
>
> | perturbation | GEARS | Linear Baseline | PerturbODE |
> |--------------|------:|----------------:|-----------:|
> | ASCL1        | 0.10  | 0.00            | 0.08       |
> | ID1          | 0.22  | 0.02            | 0.34       |
> | IRF3         | 0.36  | 0.02            | 0.34       |
> | KCNIP4       | NaN   | 0.04            | 0.38       |
> | MSX2         | 0.10  | 0.02            | 0.28       |
> | POU2AF1      | 0.26  | 0.00            | 0.30       |
> | SETDB1       | 0.04  | 0.04            | 0.20       |
> | TEAD1        | 0.24  | 0.08            | 0.30       |
> | ZBTB37       | 0.22  | 0.04            | 0.30       |
> | ZNF69        | 0.36  | 0.08            | 0.34       |
> | **median**   | **0.22** | **0.03**     | **0.30**   |
> | **std**      | **0.107** | **0.027**   | **0.091**  |
>
>
>
> ***Weakness 2: The model shows high variability and instability. Authors actually acknowledge this themselves, especially when it is tested on longitudinal time-series Perturb-seq data.***
>
> We agree that the method shows more instability on time-series data.  However, the method was primarily developed for data without additional timepoints, and those experiments were primarily meant to demonstrate that the method can be extended.  In Section 4.2, which consists of our main results on real data, we significantly outperform other methods in terms of gene regulatory network inference despite higher variability. Our error bars lie entirely above those of competing methods.
>
>
> ***Question 1: Why does GEARS still outperform PerturbODE on W2? Have you done wet-lab experiments to corroborate that your model generalizes well to unseen perturbations, cell lines, donors etc.?***
>
> GEARS depends on a prior gene pathway GO-graph and is especially designed for perturbation prediction, while PerturbODE focuses on gene regulatory network inference and does not rely on any biological priors. While we agree that more biological validation would be ideal, we believe our contribution on the computational side is significant and appropriate for ICML.

---

> > ### Author Rebuttal · Reviewer_YWg7 · 2026-04-01
> >
> > - The rebuttal helps, but the absolute F1 values still seem low, so the practical biological usefulness is still unclear to me.
> > - I appreciate the interpretability angle of your work, but I am not convinced that the paper establishes a methodological step beyond the current wave of continuous-time single-cell models. Recent ICLR papers already study related continuous-time & generative single-cell models, including scDFM [1], Cell-MNN [2], and SAVE [3]. So the contribution seems more like a careful combination of known ideas than a clear methodological jump.
> > - Claims regarding predicting the effects of unseen perturbations should be softer. The paper is strongest on GRN inference, not on unseen perturbation prediction. GEARS is also still better on W2. Additionally most recent Virtual Cell Challenge showed us that statistical or hybrid models are still very competitive solving this problem. I think the authors present an interesting generative model, but not convincing yet for biological generalization. Specifically, my concern is that in highly noisy, sparse, and weakly constrained settings, performance of deep learning model is limited less by modeling choices alone and more by the quality and of the data and the informativeness of prior assumptions. When there is a claim regarding predicting the effects of unseen perturbations, I would like to see analysis across datasets, cell lines, donors etc to accept biological conclusions.
> > - Since the core element of the paper is GRN inference, I would have liked to see comparisons against PC (or its variants) and GRNBoost2, which are strong models winning DREAM challenges.
> >
> > 1. Yu, C. et al. scDFM: Distributional Flow Matching Model for Robust Single-Cell Perturbation Prediction. ICLR 2026.
> > 2. von Bassewitz, J.-P. et al. Learning Explicit Single-Cell Dynamics Using ODE Representations. ICLR 2026
> > 3. Li, J. et al. SAVE: A Generalizable Framework for Multi-Condition Single-Cell Generation with Gene Block Attention. ICLR 2026

---

> > > ### Author Response · Authors · 2026-04-01
> > >
> > > We thank the reviewer for the additional feedback!
> > >
> > > - Thank you for pointing out GRNBoost2 and PC. We compared against GENIE3: the winner of DREAM5 challenge and we demonstrated that we outperform GENIE3 in real data. Also, GENIE3 is equivalent to GRNBoost2 when it scales up to the dataset, since GRNBoost2 is a GBM-based scalable version of GENIE3 based on the same architecture and algorithm. Previous GRN inference review literature [4] also finds GENIE3 to be among the strongest methods. We will include comparison to PC in our later draft.
> > > - We explicitly stated in the abstract that our perturbation tasks are extensions to our main GRN inference task.
> > > - We thank the reviewer for pointing out [1], [2], and [3]. Cell-MNN [2] only beats baseline with 5 PCs. It is not biologically meaningful to use only the top 5 PCs to capture all biological signals. On 50 PCs, it doesn’t beat baselines. Furthermore, Cell-MNN doesn’t capture global GRNs, which is a very different approach from PerturbODE. scDFM [1] and SAVE [3] are concurrent works made public less than 2 months before our submission.
> > >
> > >
> > >
> > > [1] Yu, C. et al. scDFM: Distributional Flow Matching Model for Robust Single-Cell Perturbation Prediction. ICLR 2026.
> > >
> > > [2] von Bassewitz, J.-P. et al. Learning Explicit Single-Cell Dynamics Using ODE Representations. ICLR 2026
> > >
> > > [3] Li, J. et al. SAVE: A Generalizable Framework for Multi-Condition Single-Cell Generation with Gene Block Attention. ICLR 2026
> > >
> > > [4] Pratapa, A., Jalihal, A. P., Law, J. N., Bharadwaj, A., and Murali, T. M. Benchmarking algorithms for gene regulatory network inference from single-cell transcriptomic data. Nature Methods, 17:147–154, 2020. doi: 10.1038/s41592-019-0690-6. URL https://doi. org/10.1038/s41592-019-0690-6.

---

### Official Review · Reviewer_SvWn · 2026-03-13

**Soundness:** 3
**Presentation:** 3
**Significance:** 3
**Originality:** 3
**Overall Recommendation:** 4
**Confidence:** 2

**Summary:**

This work discusses a central context: using perturbational single-cell transcriptomics to jointly model cellular dynamics and recover gene regulatory networks (GRNs). The authors strive to study a critical challenge by proposing PerturbODE, an interpretable neural ODE that encodes GRNs through module-structured dynamics and is evaluated on simulations and the TF Atlas overexpression dataset. The paper reports stronger large-scale GRN recovery than several causal-graph baselines and competitive held-out perturbation prediction, though some key claims still rely on somewhat limited validation and incomplete theoretical support. The biological setting is timely because TF Atlas is indeed a large perturbational resource for directed differentiation, and the baselines GEARS, scGPT, and Bicycle are relevant recent comparators in perturbation prediction / causal discovery.

**Compliance With Llm Reviewing Policy:**

Affirmed.

**Final Justification:**

addressed my main concerns

**Key Questions For Authors:**

1. The main conclusions on real data rely heavily on **incomplete reference GRNs**. Can the authors provide more systematic validation of both positive and negative edges, or report more robust calibration- or precision-oriented metrics beyond recall and p-values?

2. Thresholding and sparsity can strongly affect results. Since the thresholding strategies for PerturbODE, GENIE3, and DCDFG are not fully symmetric, can the authors provide a stricter comparison under a **matched sparsity budget** or across full **precision-recall curves**?

3. The method emphasizes “interpretable modules,” but the number of modules is an important hyperparameter. How robust are the GRN recovery results, perturbation prediction performance, and biological interpretability when the module count varies from 100 to 200 or beyond?

**Limitations:**

yes

**Strengths And Weaknesses:**

## Strengths

1. **Novel method addressing an important problem**: The paper presents a fairly unified framework that combines interpretable neural ODEs, modular representations, intervention modeling, and GRN recovery, targeting both structure discovery and unseen-perturbation prediction.

2. **Large-scale experiments are reasonably convincing**: On TF Atlas, PerturbODE* outperforms NO-TEARS / NO-TEARS-LR / DCDFG in recall and statistical significance, achieves higher AUPRC under ChIP-Atlas validation, and includes module enrichment analysis to support its interpretability.

3. **Strong biological interpretability**: The hidden neurons are interpreted as gene modules, and the paper shows correspondence with pathways such as embryonic development and angiogenesis, which is more valuable than a purely black-box dynamical model.

## Weaknesses

1. **The empirical advantage is not uniformly stable**: In some simulation settings, PerturbODE is not the best-performing method; for example, it underperforms NO-TEARS and GENIE3 on 10-gene BoolODE, and is not consistently state-of-the-art on 100/400-gene SERGIO.

2. **The evaluation protocol may be somewhat biased**: The main TF Atlas results rely heavily on recall and p-values, while the ground-truth network only contains activator edges; threshold selection, graph sparsity, and differences in the number of predicted edges across methods could substantially affect the conclusions.

---

> ### Author Rebuttal · Authors · 2026-03-30
>
> ***Weakness 1: The empirical advantage is not uniformly stable: In some simulation settings, PerturbODE is not the best-performing method; for example, it underperforms NO-TEARS and GENIE3 on 10-gene BoolODE, and is not consistently state-of-the-art on 100/400-gene SERGIO.***
>
> PerturbODE’s performance becomes more dominant on higher-dimensional real data since the inductive bias from gene modules becomes more pronounced [1]. The non-additive non-linear interaction between functionally related genes is better captured via the gene module setup. On the other hand, on SERGIO simulated data, gene regulatory relations between genes are additive (after independent Hill function activation of each gene).
>
> In the data simulated by BoolODE, the toy GRN has significantly more upstream regulators than downstream regulated genes, which is unrealistic when compared to real biological data. PerturbODE’s L1 penalty on matrix B (matrix encoding regulators) in the two-layer MLP encourages sparsity on regulators, leading to decreased performance in the toy data.
>
> ***Weakness 2: The evaluation protocol may be somewhat biased: The main TF Atlas results rely heavily on recall and p-values, while the ground-truth network only contains activator edges; threshold selection, graph sparsity, and differences in the number of predicted edges across methods could substantially affect the conclusions.***
>
> ***Question 1: The main conclusions on real data rely heavily on incomplete reference GRNs. Can the authors provide more systematic validation of both positive and negative edges, or report more robust calibration- or precision-oriented metrics beyond recall and p-values?***
>
> For the main TF Atlas, we also included comparisons with AUPRC with a reference GRN from ChiP-seq data in Figure 6. This comparison is unaffected by “threshold selection, graph sparsity, and differences in the number of predicted edges”. The reference GRN with ChiP-seq data includes activators, repressors, and negatives (no regulation). Further, in Figure 5, we show the recall score across all sparsity levels, at different thresholds. We outperform other methods in both settings, and these demonstrations shouldn’t be affected by sparsity levels.
>
> Moreover, for the thresholded results, we chose the thresholds so that most methods predict a similar number of edges. NO-TEARS and NO-TEARS-LR predicted much fewer edges since they tend to predict empty graphs (See Table 5 in the appendix).
>
>
> ***Question 2: The method emphasizes “interpretable modules,” but the number of modules is an important hyperparameter. How robust are the GRN recovery results, perturbation prediction performance, and biological interpretability when the module count varies from 100 to 200 or beyond?***
>
> As we add more modules, the performance improves up to around 350 modules. We chose 200 from the validation loss as this is a hyperparameter. Assessing the W2 validation loss would be more appropriate for hyperparameter tuning. The optimal number of modules depends on a combination of universal approximation and latent factors.
>
> We further report the inference performance with a varying number of modules (benchmarking on all 223 genes that are available in both TF Atlas and the ChIP-Atlas).
>
> | Number of Modules | AUPRC    |
> | ----------------- | -------- |
> | 10                | 0.017528 |
> | 50                | 0.015840 |
> | 100               | 0.018932 |
> | 150               | 0.016924 |
> | 200               | 0.017153 |
> | 250               | 0.017880 |
> | 300               | 0.019682 |
> | 350               | 0.022517 |
> | 400               | 0.021096 |
>
>
>
>
> [1] Segal, E., Shapira, M., Regev, A., Pe’er, D., Botstein, D., Koller, D., & Friedman, N. (2003). Module networks: Identifying regulatory modules and their condition-specific regulators from gene expression data. Nature Genetics, 34(2), 166–176. https://doi.org/10.1038/ng1165

---

> > ### Author Rebuttal · Reviewer_SvWn · 2026-04-04
> >
> > Address my concern

---

### Decision · Program_Chairs · 2026-04-30

**Decision:**

Accept (regular)

**Comment:**

This paper proposes PerturbODE, which predicts cell state trajectories using ODE under genetic perturbations and extract causal GRNs directly from the ODE parameters. The model employs a 2-layer MLP structure that implicitly represent the gene module, and is trained by minimizing the Wasserstein-2 distance between predicted and observed cell distributions. The authors evaluate GRN inference and unseen perturbation prediction on simulated dataset (SERGIO, BoolODE) and the TF Atlas dataset.

## Strengths

The motivation is solid, and the method is biologically grounded. Using ODEs to model cell state trajectories is a natural choice given the continuous nature of gene regulatory dynamics, and applying a 2-layer MLP to capture gene modules reflects well-established biological knowledge that genes are co-regulated in pathway units rather than individually. Each component of the model such as sigmoid activation mirroring the Hill function, decay term for RNA degradation, module factorization has a clear biological justification.
Scalability is another strength. Existing causal discovery methods such as Bicycle and DCDI fail beyond a few hundred genes due to memory or numerical issues, while PerturbODE scales to 812 genes on TF Atlas by factorizing the GRN into two lower-dimensional matrices.


## Weaknesses

The experimental evaluation is insufficient. The paper omits widely recognized benchmarks for GRN inference and causal discovery such as BEELINE and CausalBench. The synthetic experiments are conducted under clean conditions with perfect interventions, which closely aligns with the proposed framework’s assumptions. Evaluating on self-constructed synthetic datasets and a single real dataset does not establish generalizability.
The performance improvements are not consistent across settings. PerturbODE underperforms GENIE3 and NO-TEARS on 10-gene BoolODE, is comparable to DCDI on 100-gene SERGIO, and only shows clearer advantages at 400 genes where some baselines fail to run. On TF Atlas, GEARS still outperforms PerturbODE on W2 distance for held-out perturbation prediction. This inconsistency makes it difficult to identify the regime in which PerturbODE is reliably the better choice.
The model exhibits high variability and instability. In held-out perturbation prediction, the standard deviation of W2 is significantly larger than GEARS’, and several TFs show very large errors.
The writing has room for improvement.

While the authors addressed most reviewer concerns during the rebuttal period, the fundamental issue of insufficient experimental evaluation still remains. The evaluation is limited to self-constructed synthetic datasets and a single real dataset, and additional experiments on widely recognized, general-purpose benchmarks would be needed to establish the generalizability of the proposed method.